METHODS AND RESOURCES

## ORANGE: A CRISPR/Cas9-based genome editing toolbox for epitope tagging of endogenous proteins in neurons

Jelmer Willems [1], Arthur P. H. de Jong [1], Nicky Scheefhals [1], Eline Mertens [1], Lisa A. E. Catsburg [1], Rogier B. Poorthuis [2], Fred de Winter [3], Joost Verhaagen [3], Frank J. Meye [2], Harold D. MacGillavry [1] *

1 Cell Biology, Neurobiology and Biophysics, Department of Biology, Faculty of Science, Utrecht University, Utrecht, the Netherlands, 2 Department of Translational Neuroscience, UMC Utrecht Brain Center, Utrecht University, Utrecht, the Netherlands, 3 Laboratory for Neuroregeneration, Netherlands Institute for Neuroscience, Royal Netherlands Academy of Arts and Sciences, Amsterdam, the Netherlands

☺ These authors contributed equally to this work.
* h.d.macgillavry@uu.nl

**Data Availability Statement:** All relevant data are within the paper and its Supporting Information files.

## Abstract

The correct subcellular distribution of proteins establishes the complex morphology and function of neurons. Fluorescence microscopy techniques are invaluable to investigate subcellular protein distribution, but they suffer from the limited ability to efficiently and reliably label endogenous proteins with fluorescent probes. We developed ORANGE: Open Resource for the Application of Neuronal Genome Editing, which mediates targeted genomic integration of epitope tags in rodent dissociated neuronal culture, in organotypic slices, and in vivo. ORANGE includes a knock-in library for in-depth investigation of endogenous protein distribution, viral vectors, and a detailed two-step cloning protocol to develop knock-ins for novel targets. Using ORANGE with (live-cell) superresolution microscopy, we revealed the dynamic nanoscale organization of endogenous neurotransmitter receptors and synaptic scaffolding proteins, as well as previously uncharacterized proteins. Finally, we developed a mechanism to create multiple knock-ins in neurons, mediating multiplex imaging of endogenous proteins. Thus, ORANGE enables quantification of expression, distribution, and dynamics for virtually any protein in neurons at nanoscale resolution.

## Introduction

Neurons are highly complex cells with numerous functionally and structurally distinct subcellular compartments that are each composed of unique repertoires of molecular components. The correct targeting and localization of protein complexes and their spatial organization within subcellular domains underlies virtually every aspect of neuronal functioning. Thus, investigating the dynamic distribution of proteins in neurons is critical for a mechanistic understanding of brain function. Precise localization of individual protein species using fluorescence microscopy has become an essential technique in many fields of neuroscience and, in

**Funding:** This work was supported by the Netherlands Organization for Scientific Research (ALWOP.191 to HDM and Graduate Program of Quantitative Biology and Computational Life Sciences to NS; https://www.nwo.nl/), the European Research Council (ERC-StG 716011; https://erc.europa.eu/funding/starting-grants), and the Brain and Behavior Research Foundation (NARSAD Young Investigator Award 24995; https://www.bbrfoundation.org/grants-prizes/narsad-young-investigator-grants) to HDM. The funders had no role in study design, data collection and analysis, decision to publish, or preparation of the manuscript.

**Competing interests:** The authors have declared that no competing interests exist.

**Abbreviations:** AAV, adeno-associated virus; AMPA, α-amino-3-hydroxy-5-methyl-4-isoxazolepropionic acid; Arpc5, actin-related protein 2/3 complex subunit 5; CA1, cornu ammonis region 1; CAKE, conditional activation of knock-in expression; CaMKIIα, Calcium/calmodulin-dependent protein kinase type II subunit alpha; CAPS1, Calcium-dependent activator protein for secretion 1; $Ca_V$, voltage-dependent $Ca^{2+}$-channel; DBSCAN, density-based spatial clustering of applications with noise; DIV, day in vitro; Doc2a, double C2-like domain-containing protein a; DSB, double-stranded break; dSTORM, direct stochastic optical reconstruction microscopy; ER, endoplasmic reticulum; FLEx, flip-excision; FRAP, fluorescence recovery after photobleaching; FRRS1L, Ferric-chelate reductase 1-like protein; GFP, green fluorescent protein; GluA1, Glutamate receptor AMPA 1; GluN1, Glutamate receptor NMDA 1; gRNA, guide RNA; GSG1L, Germ cell-specific gene 1-like protein; gSTED, gated stimulated-emission depletion; HA, hemagglutinin; HDR, homology-directed repair; HITI, homology-independent targeted integration; ITR, inverted terminal repeat; Jasp, Jasplakinolide; JF646, Janelia Fluor 646; LC, light chain; KASH, Klarsicht, ANC-1, Syne Homology; LV, lentiviral; mEos3.2, monomeric Eos 3.2; MOC, Manders' overlap correlation; Munc13, mammalian uncoordinated 13; NHEJ, nonhomologous end joining; Nlgn3, neuroligin 3; NMDA, *N*-methyl-D-aspartate; ORANGE, Open Resource for the Application of Neuronal Genome Editing; PALM, photoactivated localization microscopy; PAM, protospacer adjacent motif; PCC, Pearson's correlation coefficient; PSD95, postsynaptic density protein 95; Rab, ras-related protein; RFP, red fluorescent protein; RIM1, Rab3-interacting molecule 1; RNP, ribonucleoprotein; ROI, region of interest; SHANK, SH3 and multiple ankyrin repeat

particular, for studies on synaptic function, in which protein mislocalization at scales less than 1 μm can already significantly affect synaptic efficacy [1]. Recently developed superresolution imaging methods now routinely achieve spatial resolution as low as tens of nanometers, allowing determination of protein distributions at the molecular scale [2,3]. Consequently, these methods are highly sensitive to experimental alterations that affect protein organization, and efficient labeling techniques that accurately report the localization of endogenous proteins are critical.

Visualization of subcellular protein localization typically relies on antibody-based labeling approaches or overexpression of fluorescently tagged proteins, but both techniques have serious limitations [4]. Immunostaining largely relies on the availability of specific antibodies, which has severely hampered progress for many targets. Immunostaining also precludes labeling and visualization of protein dynamics in live cells, and penetration of antibodies in thick tissue samples is challenging. Additionally, it is often desirable to sparsely label individual cells to measure protein distribution at high contrast, which is difficult to achieve with immunostaining-based techniques. Expression of fluorescently tagged proteins overcomes many of these issues; however, exogenous expression of recombinant proteins can lead to mislocalization and can induce severe morphological and/or physiological artifacts. For instance, overexpression of synaptic proteins such as postsynaptic density protein 95 (PSD95) and SH3 and multiple ankyrin repeat domains protein (Shank) have pronounced effects on synapse number, content, structure, and physiology [5–8]. Exorbitant expression levels can be circumvented by a replacement strategy in which a tagged protein is expressed in a knock-down or knock-out background [9], but this will, at best, only approximate endogenous levels and is uncoupled from endogenous transcriptional or translational regulatory mechanisms. Recombinant antibody-based approaches have been developed for live-cell imaging of neuronal proteins, but they have so far been restricted to a few targets [10–14]. The generation of fluorescently tagged knock-ins (for instance, in mouse lines) prevents these issues. However, the generation and maintenance of transgenic animals is costly and time consuming, making it an inefficient approach for high-throughput tagging of neuronal proteins. Also, generally, transgenic labeling leads to expression of tagged proteins in all cells, thus limiting the options for imaging in individual cells.

In view of the limitations of current techniques, we sought to develop a protein labeling strategy that meets the following criteria: (1) accurately reports a single protein species at endogenous protein levels and with spatiotemporal expression pattern; (2) can be rapidly developed and expanded to many proteins of interest; (3) does not interfere with protein localization and function; (4) can be applied in dissociated neuronal cultures, organotypic slice culture, and in vivo; (5) allows for sparse labeling of neurons; and (6) is compatible with (superresolution) light microscopy of live as well as fixed tissues. We reasoned that labeling of endogenous proteins with fluorescent tags using targeted gene-editing techniques would fulfill all these criteria.

Targeted gene editing using CRISPR/Cas9 facilitates the introduction of donor DNA at specific loci in the genome, effectively tagging endogenous proteins of interest [15,16]. For neuronal cells, several CRISPR/Cas9-based knock-in strategies have been developed, relying on different mechanisms to repair double-stranded breaks (DSBs) introduced by Cas9. One strategy is based on homology-directed repair (HDR) to insert donor DNA into the genomic locus [17,18]. However, HDR preferentially occurs during the S/G2 phases of the cell cycle and is significantly down-regulated in postmitotic cells [19]. This limits the application of HDR in neurons, although successful integration can still be observed with highly elevated donor DNA levels or via a combination of donor cleavage and microhomology arms [20,21]. Additionally, in order to be efficient, HDR requires the addition of long homology arms to the donor DNA,

domains protein; SMLM, single-molecule localization microscopy; SO, stratum oriens; SP, stratum pyramidale; SR, stratum radiatum; SpCas9, *Streptococcus pyrogenes* Cas9; Syt7, Synaptotagmin 7; TARP, Transmembrane AMPAR regulatory protein; Term, termination sequence; uPAINT, universal point accumulation in nanoscale topography; WASP1, Wiskott-Aldrich syndrome protein 1.

which can be laborious to generate, considerably complicating the development of knock-in constructs.

Alternative strategies are based on nonhomologous end joining (NHEJ) to repair DSBs, which is active throughout the cell cycle, as well as in postmitotic cells, and can be used to insert donor DNA with high efficiency [22–24]. Based on NHEJ, the homology-independent targeted integration (HITI) method for endogenous protein tagging in postmitotic neurons was previously developed and outperformed HDR-based methods [21,24]. We hypothesized that HITI would provide an accessible and scalable approach for the tagging of endogenous proteins in neurons, in dissociated neuronal cultures and organotypic cultures, and in vivo. However, applications of this method have so far been limited to a few target proteins [21,24–26]. In addition, designing and cloning of knock-in constructs, the compatibility of DNA delivery methods for various tissue preparations, and validation of NHEJ-based knock-in accuracy and efficiency have until now been quite challenging and have not been addressed systematically.

Here, based on HITI, we developed ORANGE, an Open Resource for the Application of Neuronal Genome Editing, which offers researchers the means to endogenously tag proteins of interest in neurons, allowing for the accurate investigation of protein expression, localization, and dynamics. This toolbox includes (1) a single template vector that contains the complete knock-in cassette, which can be adapted in two straightforward cloning steps to tag virtually any protein of interest, and (2) a library of readily usable knock-in constructs targeting a set of 38 proteins. This library encompasses a wide variety of proteins, including cytoskeletal components, signaling molecules, endosomal markers, presynaptic and postsynaptic scaffolds, adhesion complexes, and receptors. We show that this tagging strategy facilitates protein labeling in dissociated neuronal culture, in organotypic slice cultures, and in vivo with high accuracy and without overexpression artifacts. Moreover, we demonstrate that this toolbox facilitates live-cell and superresolution imaging of endogenous proteins to resolve their localization and dynamics in neurons at high spatial and temporal resolution. We furthermore show that ORANGE can be combined with the Cre-Lox system driving the conditional expression of genetically encoded reporters. Finally, we developed a Cre-dependent knock-in strategy for multiplex labeling of proteins within single cells. Altogether, we present a robust and easy-to-implement toolbox for the tagging and visualization of endogenous proteins in postmitotic neurons, allowing for in-depth investigation of diverse neuronal cell biological processes.

## Results

### ORANGE knock-in toolbox to fluorescently tag endogenous proteins in neurons

We first aimed to design a simple workflow to facilitate the rapid generation of knock-in constructs using conventional molecular cloning approaches. To this end, we designed a single CRISPR/Cas9 knock-in template vector (pORANGE) based on the original NHEJ-mediated HITI method [24]. Our design allows for the flexible insertion of a unique 20-nucleotide target sequence that guides Cas9 to the genomic locus of interest and a donor sequence containing the knock-in sequence (e.g., green fluorescent protein [GFP]) that will be inserted in the targeted genomic locus (Fig 1A and S1 Fig). The generated knock-in construct contains all elements required for targeted CRISPR/Cas9-based genome editing: (1) a U6-driven expression cassette for the guide RNA (gRNA) targeting the genomic locus of interest, (2) the donor sequence containing the (fluorescent) tag, and (3) a Cas9 expression cassette driven by a universal β-actin promoter (Fig 1A). The donor sequence is generated by standard PCR, with

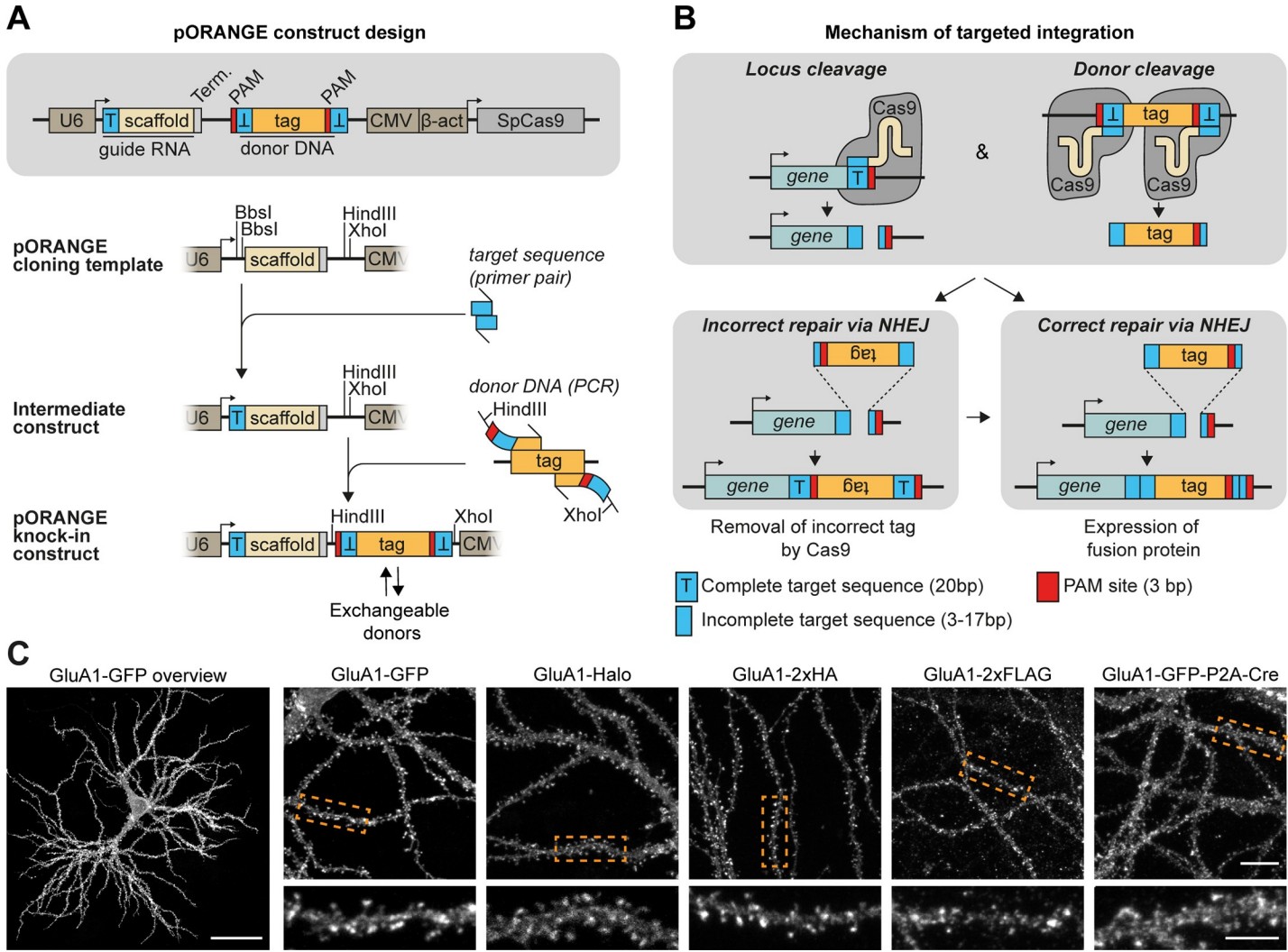

**Fig 1. ORANGE: An easy-to-implement toolbox for endogenous tagging of proteins in neurons.** (A) Overview of the pORANGE knock-in construct. To clone knock-in constructs, first a 20-bp target sequence for the genomic locus of interest is ligated in the guide RNA cassette. Then, the donor sequence containing the tag of interest is generated by PCR and inserted in the donor DNA cassette. (B) Mechanism of ORANGE-mediated gene targeting. (C) Examples of knock-in neurons expressing GluA1 tagged with GFP, HaloTag, small epitope tags (2× HA, 2× FLAG), or GFP-P2A-Cre recombinase. Dashed boxes indicate zooms. Scale bars, 40 μm for the GluA1-GFP overview (far left), 10 μm for individual overviews, and 5 μm for the zooms. β-act, β-actin; CMV, cytomegalovirus; GFP, green fluorescent protein; GluA, Glutamate receptor AMPA 1; HA, hemagglutinin; NHEJ, nonhomologous end joining; ORANGE, Open Resource for the Application of Neuronal Genome Editing; PAM, protospacer adjacent motif; SpCas9, *Streptococcus pyrogenes* Cas9; T, target sequence; Term, termination sequence.

primers introducing a short linker and Cas9 target sequences flanking the donor (Fig 1A). These target sequences are identical to the genomic target sequence. As a result, the gRNA used to create a genomic DSB is also used to remove the donor DNA from the plasmid, allowing for its genomic integration. Importantly, the orientation of the target sequence and protospacer adjacent motif (PAM) sites flanking the donor is inverted compared with the genomic sequence to guarantee that integration occurs in the correct orientation (Fig 1B). For a detailed description of genomic target sequence selection, gRNA sequence, and donor PCR primer design, we refer to the design and cloning protocol in the Materials and methods section (also see S1 Fig). This approach is flexible because the donor sequence can be easily exchanged for different fluorophores, self-labeling enzymes like Halo, small epitope tags like hemagglutinin (HA) and FLAG, or larger donors like GFP-P2A-Cre to meet the specific demands for the

experiment (Fig 1C). We found that this cloning strategy is easy to employ and enables the rapid and flexible generation of knock-in constructs.

Using the pORANGE template vector, we designed and generated a library providing knock-in constructs to endogenously label 38 proteins for fluorescence imaging (Fig 2, S2 Fig,

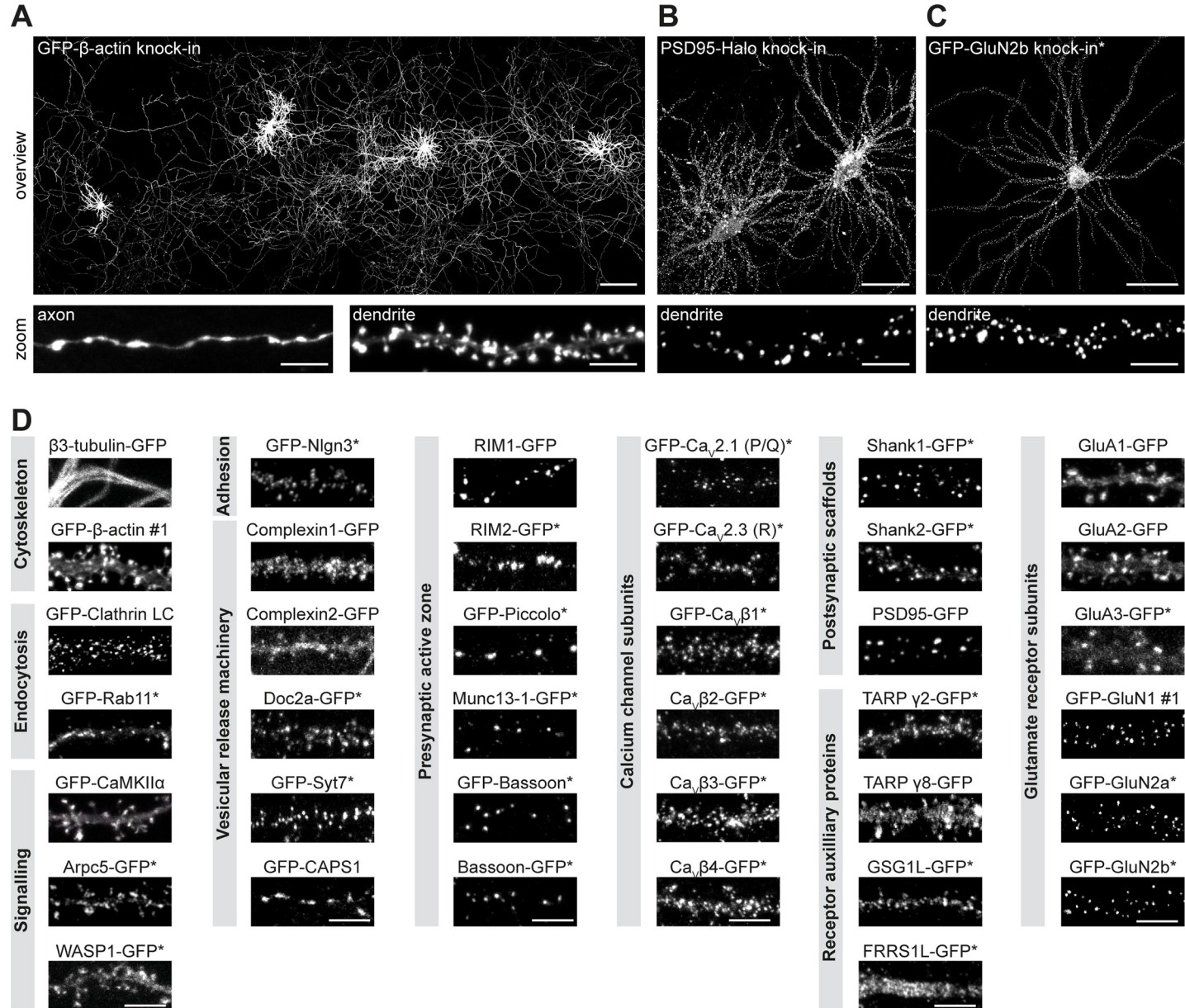

**Fig 2. A versatile ORANGE knock-in library for endogenous tagging of proteins in neurons.** (A) Example at low magnification showing four GFP-β-actin knock-in neurons (DIV 21). Zooms show an axon and dendrite, respectively. Scale bars: overview, 200 μm; zoom, 5 μm. (B) Example of two PSD95-Halo knock-in neurons (DIV 21). Zoom shows a single dendrite. Scale bars: overview, 40 μm; zoom, 5 μm. (C) Example of GFP-GluN2b knock-in neuron (DIV 21). Scale bars: overview, 40 μm; zoom, 5 μm. (D) Representative images of ORANGE knock-in neurons, categorized according to protein function or subcellular localization. Neurons were transfected at DIV 3 and imaged at DIV 21. Scale bars, 5 μm. Asterisk indicates signal enhancement with anti-GFP staining (Alexa488 or Alexa647). Arpc5, actin-related protein 2/3 complex subunit 5; CaMKIIα, Calcium/calmodulin-dependent protein kinase type II subunit alpha; CAPS1, Calcium-dependent activator protein for secretion 1; $Ca_V$, voltage-dependent $Ca^{2+}$-channel; DIV, day in vitro; Doc2a, double C2-like domain-containing protein a; FRRS1L, Ferric-chelate reductase 1-like protein; GFP, green fluorescent protein; GluA, glutamate receptor AMPA 1; GluN, Glutamate receptor NMDA 1; GSG1L, Germ cell-specific gene 1-like protein; LC, light chain; Munc13, mammalian uncoordinated 13; Nlgn3, neuroligin 3; ORANGE, Open Resource for the Application of Neuronal Genome Editing; PSD95, postsynaptic protein 95; Rab11, ras-related protein 11; RIM: Rab3-interacting molecule; Shank, SH3 and multiple ankyrin repeat domains protein; Syt7, Synaptotagmin 7; TARP, Transmembrane AMPAR regulatory protein; WASP1, Wiskott-Aldrich syndrome protein 1.

and S5 Table). To cover the many areas of neuronal cell biology, we selected proteins representing various molecular processes, including cytoskeletal components, intracellular signaling molecules, trafficking proteins, synaptic scaffolds, and receptor subunits. We were able to directly image the fluorescent GFP signal for many endogenously tagged proteins. However, for less abundant proteins (e.g., calcium channel subunits, presynaptic active zone proteins, and *N*-methyl-D-aspertate (NMDA) receptor subunits), amplification of the GFP tag with anti-GFP antibodies was required to visualize protein distribution (indicated with an asterisk in Fig 2 and S2 Fig). Throughout this study, we refer to knock-in constructs as the name of the protein that is labeled: in N-terminally tagged proteins, the tag is in front of the protein name, and in C-terminally tagged proteins, the tag is after.

For several proteins in our library, no specific antibodies are available. In order to compare their subcellular distribution to what is reported in literature (S1 Table), we costained several knock-ins with pre- or postsynaptic markers and confirmed the expected distribution for all of the constructs we evaluated (S3 Fig). Together, our ORANGE toolbox includes a broad library of knock-in constructs and provides an efficient strategy to adapt or design constructs with relative ease using standardized cloning techniques.

## Viral delivery of ORANGE to label endogenous proteins in dissociated neuronal cultures and organotypic slice cultures and in vivo

Adeno-associated virus (AAV)-based DNA delivery has become a valuable method of administration, especially for in vivo applications. To test whether this approach is compatible with ORANGE, we generated HaloTag knock-in constructs for PSD95 and Glutamate receptor AMPA 1 (GluA1) and subcloned these into AAV vectors. AAVs were injected in the cornu ammonis region 1 (CA1) of the hippocampus in Cas9-P2A-GFP transgenic mice [27] (Fig 3A). After 4 weeks, acute slices were prepared and live-stained with Halo-JF646. This resulted in fast labeling deep into the tissue. For both PSD95 and GluA1, efficient knock-in labeling was observed in CA1, as well as in CA3 and subiculum, with additional labeling in the dentate gyrus (Fig 3C and 3D). At higher magnifications, we only observed neurons with punctate, synaptic expression of PSD95-Halo. Similarly, although to a lesser extent, GluA1-Halo was also highly enriched in dendritic spines, as expected (Fig 3E and 3G). Finally, we used superresolution gated stimulated-emission depletion (gSTED) imaging to resolve individual synapses at high resolution (Fig 3F and 3H).

Next, we tested whether ORANGE knock-ins could also be delivered using lentiviral (LV) vectors. We divided the ORANGE knock-in cassette over two LV constructs (S4A Fig) because the full cassette exceeds the packaging limit of LV particles. Also, premature coexpression of Cas9 and the gRNA during the production of viral particles in packaging cells would lead to removal of the donor DNA. Both in dissociated hippocampal cultures and in organotypic slice cultures, we observed knock-ins, showing that LVs can be used to successfully express ORANGE knock-ins (S4 Fig). Together, these results show that ORANGE is compatible with various modes of DNA delivery suitable for labeling in dissociated neuronal cultures, in organotypic slice cultures and in vivo, broadening the potential applications of this CRISPR/Cas9 genome editing toolbox.

## ORANGE enables fast and accurate donor integration

To test the rate of donor integration and subsequent expression of tagged proteins with lipofection-based DNA delivery, we cotransfected dissociated hippocampal neurons at day in vitro (DIV) 3 with a β3-tubulin-GFP knock-in construct and a construct for soluble mCherry expression (S5 Fig). Because of the high protein turnover rate of β3-tubulin, integration of the

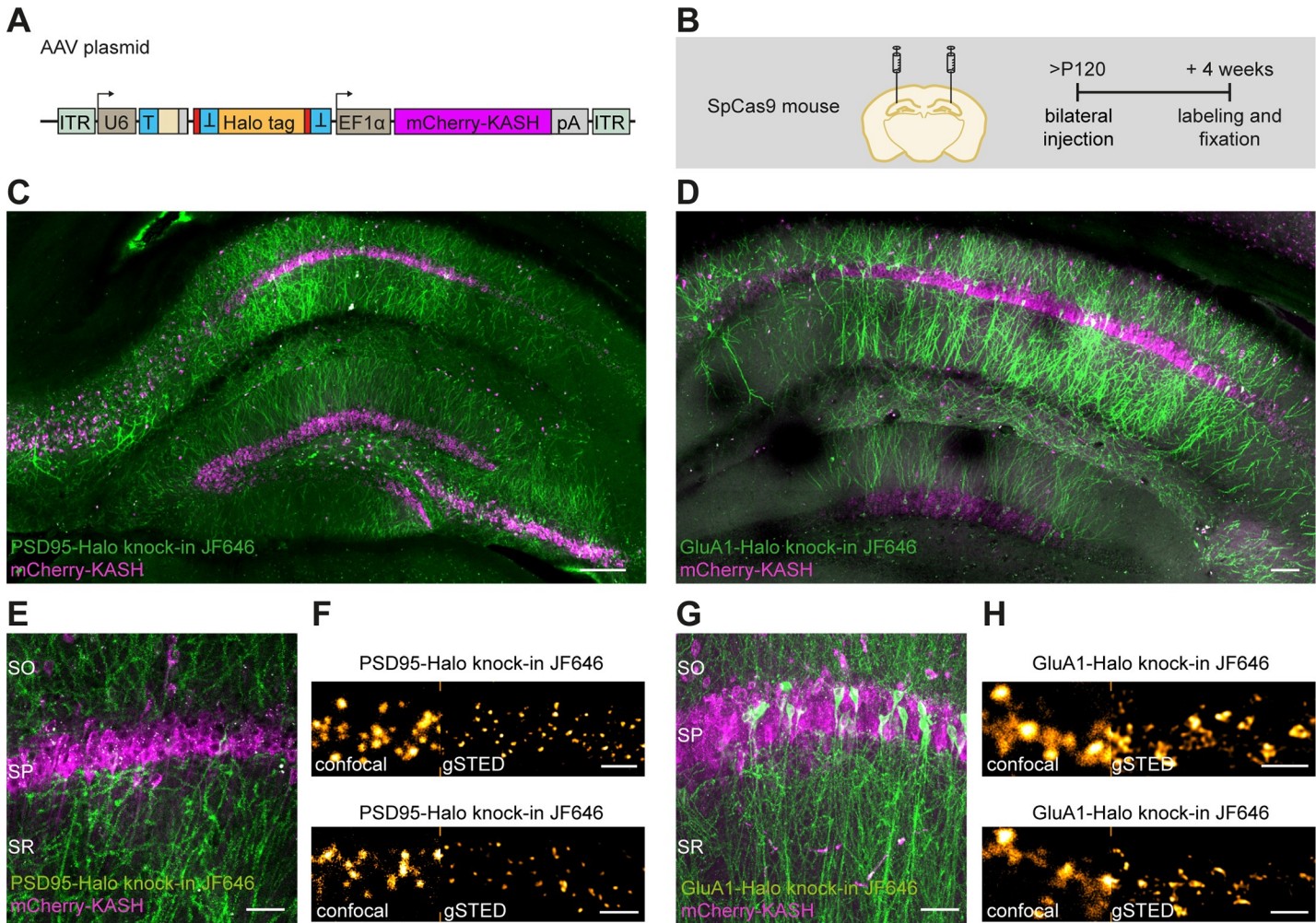

**Fig 3. ORANGE mediates in vivo genome editing.** (A) Overview of ORANGE AAV plasmid. (B) Workflow and time line for in vivo genome editing. (C and D) Confocal images of acute slices from SpCas9 mouse hippocampus injected with PSD95-Halo knock-in (C) and GluA1-Halo knock-in (D) AAV vectors visualized with Halo-JF646 ligand (green). Infected cells are positive for mCherry-KASH (magenta). Scale bar, 100 μm. (E and G) Zooms for acute slices as shown in (C) and (D), respectively. Scale bar, 40 μm. (F and H) Representative images of confocal and gSTED microscopy in acute slices. Shown are dendrites positive for PSD95-Halo (F) and GluA1-Halo knock-in (H). Scale bar, 2 μm. AAV, adeno-associated virus; EF1α, elongation factor 1α; GluA, Glutamate receptor AMPA; gSTED, gated stimulated-emission depletion; ITR, inverted terminal repeat; JF646, Janelia Fluor 646; KASH, Klarsicht, ANC-1, Syne Homology; ORANGE, Open Resource for the Application of Neuronal Genome Editing; pA, polyadenylation; PSD95, postsynaptic protein 95; SO, stratum oriens; SP, stratum pyramidale; SpCas9, *S. pyrogenes* Cas9; SR, stratum radiatum; T, target sequence.

donor should be rapidly observable by expression of the tagged protein. Successful labeling of β3-tubulin was observed within 24 hours after transfection, albeit at relatively low efficiency (1.1% ± 0.2% β3-tubulin GFP$^+$/mCherry$^+$ double positive cells). Labeling efficiency increased 10-fold over time and reached a plateau around 96 hours after transfection (10.9% ± 0.1% β3-tubulinGFP$^+$/mCherry$^+$; S5 Fig), indicating that donor integration preferentially takes place within the first days after transfection.

Next, we determined the accuracy of genomic integration for our knock-in library using confocal microscopy. Expression patterns were in line with available literature (S1 Table), and we did not observe aberrant or diffuse expression of the integrated tag for any of the knock-in constructs in our library. This indicates that off-target integration, or unintended GFP expression directly from the knock-in plasmid, did not occur or is extremely rare (see Discussion).

To get a detailed overview of the precision of donor integration into the targeted genomic locus, we analyzed the genomic sequence after integration for 28 GFP knock-in constructs using next-generation sequencing (S6 Fig). We detected a high frequency of in-frame integration of the GFP tag in the targeted locus for almost all knock-ins (S6B and S6C Fig). Besides correct integration, we found various insertions and deletions leading to frameshift mutations (S6B and S6D Fig). We noted that the frequency of indels was variable between different knock-ins, which is likely due to the difference in target sequences, which has been reported to highly determine the accuracy of Cas9-mediated cleavage and NHEJ-mediated repair [28,29]. Notably, the accuracy of donor integration did not correlate with the Doench on-target score [30] (Pearson r: −0.15, $R^2$: 0.02, $P < 0.05$) or Bae out-of-frame score [31] (Pearson r: 0.25, $R^2$: 0.06, $P > 0.05$) (S6E and S6F Fig). In conclusion, although out-of-frame integration occurs at varying frequencies as shown by next-generation sequencing, imaging of our knock-in library suggests that this does not result in a fluorescent signal or aberrant protein expression.

We noted that several of the knock-in constructs with lower in-frame integration, such as GFP-β-actin and GFP-Glutamate receptor NMDA 1 (GluN1), also had a low efficiency of knock-in expression in cultured neurons. To test whether this is gene specific or guide sequence specific, and in an attempt to generate more efficient knock-in constructs for these genes, we designed extra knock-in constructs for β-actin (GFP-β-actin #2) and GluN1 (GFP-GluN1 #2 and #3) by making use of alternative PAM sites (S7 Fig). All alternative PAM site variants resulted in successful GFP knock-ins in cultured neurons, with expected GFP expression patterns. Again, we did not observe neurons with aberrant distribution of GFP signal. For both genes, we found that various PAM sites along the same genomic region varied widely in their knock-in efficiency relative to the number of transfected neurons (GFP-β-actin knock-in #1: 0.42% ± 0.09%, #2: 7.4% ± 1.1%, Student t test, $P < 0.05$; GFP-GluN1 knock-in #1: 0.43% ± %0.04, #2: 3.0% ± %0.7, #3: 5.6% ± %0.4, ANOVA, $P < 0.05$). These results show that knock-in efficiency is highly dependent on the target site used for integration.

## ORANGE reliably labels proteins without overexpression artifacts

To further determine whether the integrated fluorescent tag reliably labels the endogenous target protein, we compared the localization of several knock-ins with specific antibody staining in confocal microscopy. First, we tested the knock-in construct for PSD95, a core postsynaptic scaffold molecule [32] (Fig 4). We transfected dissociated hippocampal cultures with the PSD95-GFP knock-in construct well before synaptogenesis (DIV 3) and fixed the neurons at a mature stage (DIV 21) (Fig 4A–4G). In all neurons with a detectable GFP signal, the GFP signal was found in a punctate pattern enriched in dendritic spines, characteristic for endogenous PSD95 expression. The GFP signal closely colocalized with immunolabeled PSD95 and showed a strong correlation with intensity of PSD95 immunostaining in PSD95-GFP knock-in neurons (Pearson r: 0.72, $R^2$: 0.51, $P < 0.001$, n = 550 synapses from 11 neurons; Fig 4B). To test whether the knock-in affects total PSD95 levels, we used the PSD95 antibody staining to compare protein levels between PSD95-GFP knock-in and control neurons that were transfected with soluble GFP (GFP control). Although we observed that, in a subpopulation of PSD95 knock-in neurons, protein levels were modestly lower, PSD95 levels in PSD95-GFP knock-in neurons (relative fluorescence intensity: 0.84 ± 0.04, n = 17 neurons) were on average comparable to GFP control neurons (0.98 ± 0.02, n = 15 neurons, ANOVA, $P > 0.05$) (Fig 4C; inset). In contrast, overexpression of PSD95-GFP significantly increased synaptic PSD95 protein levels (relative fluorescence intensity: 4.2 ± 0.4, n = 17 neurons, $P < 0.001$). Moreover, synapse size was significantly increased in neurons overexpressing PSD95 (0.18 ± 0.01 μm$^2$) compared with GFP control neurons (0.13 ± 0.01 μm$^2$, ANOVA, $P < 0.001$) but was

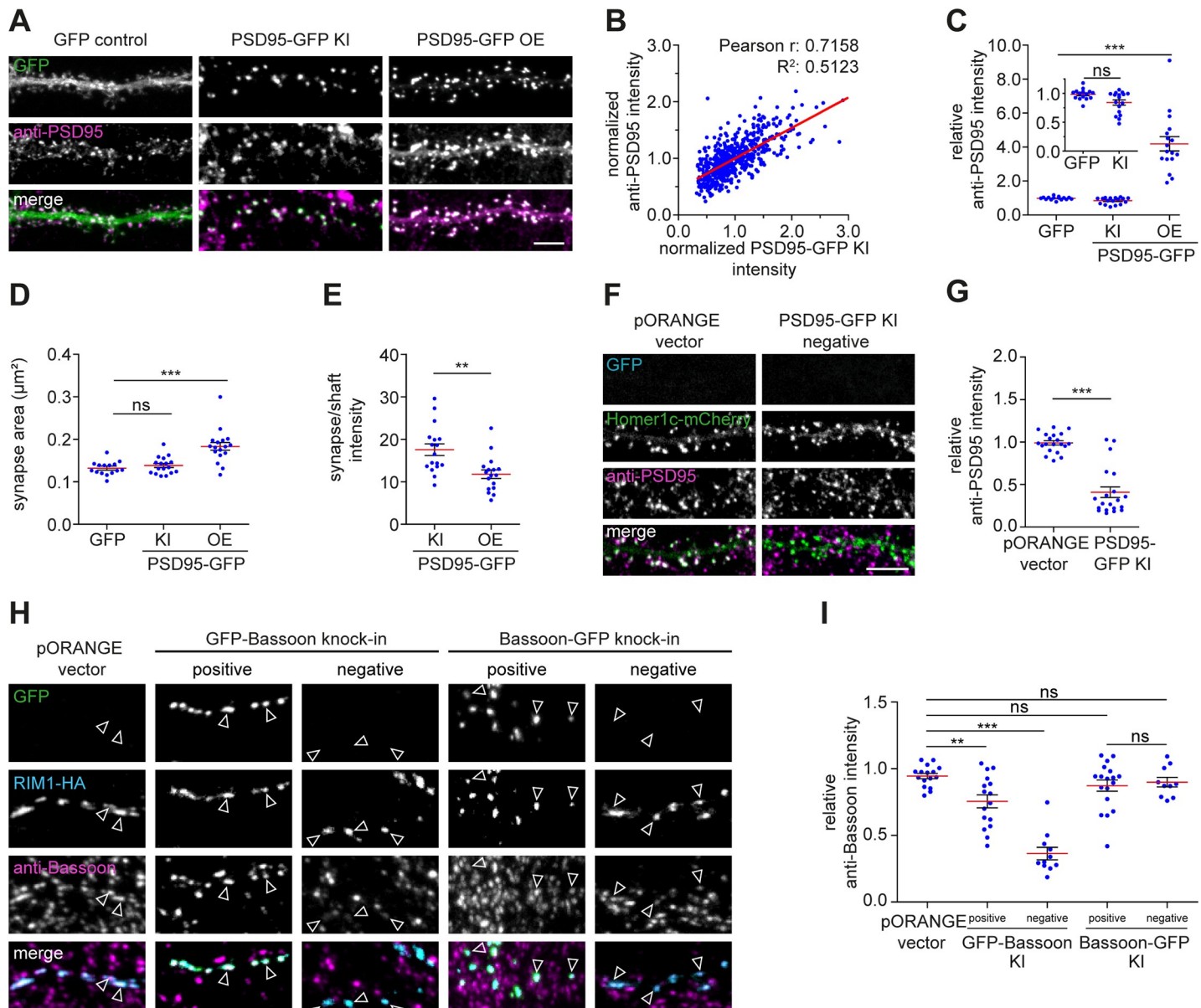

**Fig 4. Validation of ORANGE labeling efficiency.** (A) Representative images of dendrites transfected with soluble GFP, PSD95-GFP knock-in (KI) construct, or a PSD95-GFP overexpression construct (green) stained with anti-PSD95 (magenta, Alexa568). DIV 21. Scale bar, 5 μm. (B) Correlation between PSD95-GFP KI and anti-PSD95 staining intensity. (C) Quantification of synaptic PSD95 levels, (D) synapse area, and I PSD95 synapse/shaft intensity. (F) Representative images of dendrites coexpressing Homer1c-mCherry (green) and either the empty pORANGE template vector or PSD95-GFP KI construct (blue) stained with anti-PSD95 (magenta, Alexa647). DIV 21. Scale bar, 5 μm. (G) Quantification of PSD95 levels in transfected but KI-negative neurons. Data are represented as means ± SEM. * $P < 0.05$, ** $P < 0.01$, *** $P < 0.001$, ANOVA or Student $t$ test. Underlying data can be found in S1 Data. DIV, day in vitro; GFP, green fluorescent protein; HA, hemagglutinin; KI, knock-in; ns, not significant; OE, overexpression; ORANGE, Open Resource for the Application of Neuronal Genome Editing; PSD95, postsynaptic protein 95; RIM1, Rab3-interacting molecule 1.

unaffected in PSD95-GFP knock-in neurons (0.14 ± 0.001 μm², $P > 0.05$; Fig 4D). Lastly, we found that PSD95 was significantly more enriched at synapses in PSD95 knock-in cells (ratio synapse/shaft intensity: 17.6 ± 1.4) compared with PSD95-overexpressing neurons (11.8 ± 1.0, Student $t$ test, $P < 0.01$; Fig 3E), indicating that a large fraction of overexpressed PSD95 mislocalized to the dendritic shaft.

Transfection of knock-in constructs did not always result in successful knock-in of GFP (S5–S7 Figs). To determine whether in transfected but GFP knock-in-negative neurons integration of the GFP tag was simply not successful or integration introduced indels affecting protein expression, we cotransfected neurons with the PSD95-GFP knock-in construct with a Homer1c-mCherry overexpression construct to label synapses. We measured PSD95 levels in Homer1c-mCherry-positive neurons that did not show detectable PSD95-GFP signal (Fig 4F and 4G). In most of these GFP-negative neurons, PSD95 protein levels were significantly down-regulated (relative fluorescence intensity: $0.41 \pm 0.06$, $n = 20$ neurons) compared with neurons cotransfected with the empty pORANGE template vector and Homer1c-mCherry ($0.99 \pm 0.02$, $n = 20$ neurons, Student $t$ test, $P < 0.001$), suggesting partial or complete knock-out of the target protein in transfected but knock-in-negative neurons.

In addition to PSD95, we measured protein levels for several other proteins, including Shank2, Calcium/calmodulin-dependent protein kinase type II subunit alpha (CaMKIIα), β-actin (S8 Fig), and the presynaptic active zone protein Bassoon (Fig 4H and 4I) in successful knock-in neurons as well as in knock-in-negative neurons. Additionally, taking advantage of the fact that Bassoon tolerates both N-terminal and C-terminal tagging (S1 Table) [33], we designed GFP-Bassoon and Bassoon-GFP knock-in constructs to compare the effect of tagging the same protein at different positions in the gene. Both N-terminal and C-terminal Bassoon knock-ins showed an identical, punctate expression pattern and colocalized with coexpressed Rab3-interacting molecule 1 (RIM1)-HA, a presynaptic marker (Fig 4H). This indicates that, for Bassoon, endogenous tagging either at the N-terminus or C-terminus does not interfere with protein localization. Using a specific Bassoon antibody, we found that, like PSD95, most knock-in neurons express Bassoon at endogenous levels. However, the N-terminal-tagged knock-in neurons showed a slightly larger fraction of neurons with reduced levels of Bassoon (relative fluorescence intensity: $0.75 \pm 0.05$, $n = 16$ neurons, $P < 0.01$) compared with C-terminal-tagged ($0.87 \pm 0.04$, $n = 18$ neurons, $P > 0.05$) and control cells ($0.94 \pm 0.01$, $n = 16$ neurons). Notably, neurons transfected with (but negative for) the GFP-Bassoon knock-in showed significantly reduced levels of Bassoon ($0.36 \pm 0.04$, $n = 11$ neurons, $P < 0.001$), whereas transfection of the Bassoon-GFP knock-in did not affect protein levels in knock-in-negative neurons ($0.90 \pm 0.04$, $n = 10$ neurons, ANOVA, $P < 0.05$). Furthermore, we found that the GFP signal of the Shank2, CaMKIIα, and β-actin knock-ins approximated endogenous levels but that the protein levels in knock-in-negative cells varied between constructs (S8 Fig). Thus, although a successful knock-in results in accurate detection of endogenously tagged proteins, erroneous integration may lead to partial knock-out of the targeted gene in knock-in-negative neurons depending on the protein and position of integration. Altogether, these data demonstrate that ORANGE enables successful integration of fluorescent tags at the targeted genomic locus, resulting in expression of fusion proteins, which reliably reports the localization of proteins of interest, without overexpression artifacts.

## Live-cell imaging of endogenous protein dynamics

In addition to imaging fixed cells, the introduction of fluorescent tags allows for imaging of endogenous protein dynamics in living cells. To demonstrate this directly, we performed live-cell imaging on GFP-β-actin knock-in neurons. First, we confirmed that N-terminal tagging of endogenous β-actin with GFP did not alter the actin network based on phalloidin staining of fixed neurons (S8 Fig). Second, we acquired time-lapse images of GFP-β-actin knock-in neurons showing the characteristic dynamic behavior of actin in dendritic spines [34,35] (Fig 5A). Jasplakinolide (Jasp), which stabilizes actin filaments and promotes actin polymerization, rapidly reduced dendritic spine dynamics (as measured by an increase in frame-to-frame

correlation, 0.89 ± 0.01, $n$ = 7 neurons) compared with DMSO control (0.84 ± 0.01, $n$ = 7 neurons, Student $t$ test, $P < 0.01$) (Fig 5B and 5C). We noted that the diffuse actin signal was depleted from the dendritic shafts after Jasp application, indicating that the enhanced actin polymerization incorporated free actin monomers from the dendritic shaft. We further evaluated this with a fluorescence recovery after photobleaching (FRAP) assay (Fig 5D). In control neurons, β-actin turnover was fast, with a large mobile pool (mobile fraction: 0.87 ± 0.01, $n$ = 13 neurons, Fig 5E and 5F), consistent with measures based on overexpressed β-actin [36]. As expected, addition of Jasp largely abolished turnover of spine β-actin (mobile fraction: 0.02 ± 0.01, $n$ = 13 neurons, Student $t$ test, $P < 0.001$), indicating that Jasp induced integration of most GFP-β-actin in stable actin filaments. These experiments show that ORANGE knock-ins are compatible with live-cell imaging of endogenous protein dynamics in neurons.

Single-molecule imaging is a powerful tool to investigate the dynamics of individual molecules within living cells. We designed knock-in constructs targeting CaMKIIα, an abundant neuronal $Ca^{2+}$-activated signaling protein essential for learning and memory [37]. Confocal microscopy showed that the GFP-CaMKIIα knock-in was primarily cytoplasmic with moderate enrichment in spines (Fig 2 and S2 Fig), consistent with previous studies [18,38]. We next replaced GFP for monomeric Eos 3.2 (mEos3.2), a photoconvertible protein compatible with single-molecule tracking based on photoactivated localization microscopy (PALM) [39,40] (S9A Fig). Individual mEos3.2-CaMKIIα molecules were imaged to reconstruct a superresolved image of CaMKIIα distribution (S9B Fig) and to map single-molecule trajectories in spines and dendrites over time (S9C and S9D Fig). From the trajectories, we calculated the mean-squared displacements (MSDs) to derive the diffusion coefficient for individual trajectories (S9E and S9F Fig), revealing two distinct dynamic CaMKIIα populations: a mobile population (mean diffusion coefficient 0.145 ± 0.049 μm²/s) and less-mobile population (0.0140 ± 0.0011 μm²/s, $n$ = 11 neurons). Thus, genetic tagging with photoconvertible molecules such as mEos3.2 can be used for live-cell single-molecule tracking PALM experiments to resolve the distribution and dynamics of endogenous, intracellular proteins.

## Superresolution imaging of endogenously expressed proteins in neurons

We envisioned that tagging endogenous proteins particularly presents advantages for superresolution imaging by facilitating labeling of proteins in a subset of neurons and overcoming many artifacts associated with antibody specificity or overexpression of recombinant proteins. Also, this combination would be interesting for studying recently identified proteins with unknown subcellular distribution.

First, we employed our GFP-β-actin and β3-tubulin-GFP knock-in constructs to resolve and correlate their well-known subcellular organization in individual neurons using gSTED microscopy. Recent superresolution studies have demonstrated that the actin cytoskeleton forms ring-like structures that are periodically organized along axons as well as dendrites [41–43]. We tested whether we could resolve this particular organization of the actin cytoskeleton in individual β-actin knock-in neurons within dense, mature, dissociated hippocampal cultures. Using gSTED imaging, we observed distinct periodic actin structures in both the axon and dendrites (Fig 6A–6E). In addition to resolving the actin network, we performed two-color gSTED imaging of β3-tubulin-GFP knock-in neurons immunolabeled with anti-α-tubulin to resolve the neuronal microtubule network (S10A–S10C Fig). Thus, ORANGE combined with superresolution imaging is an easily accessible approach to resolve the subcellular distribution of endogenous proteins with high resolution.

Second, we took advantage of ORANGE to perform two-color gSTED imaging on synaptic proteins. To assess the performance of ORANGE knock-ins compared with conventional

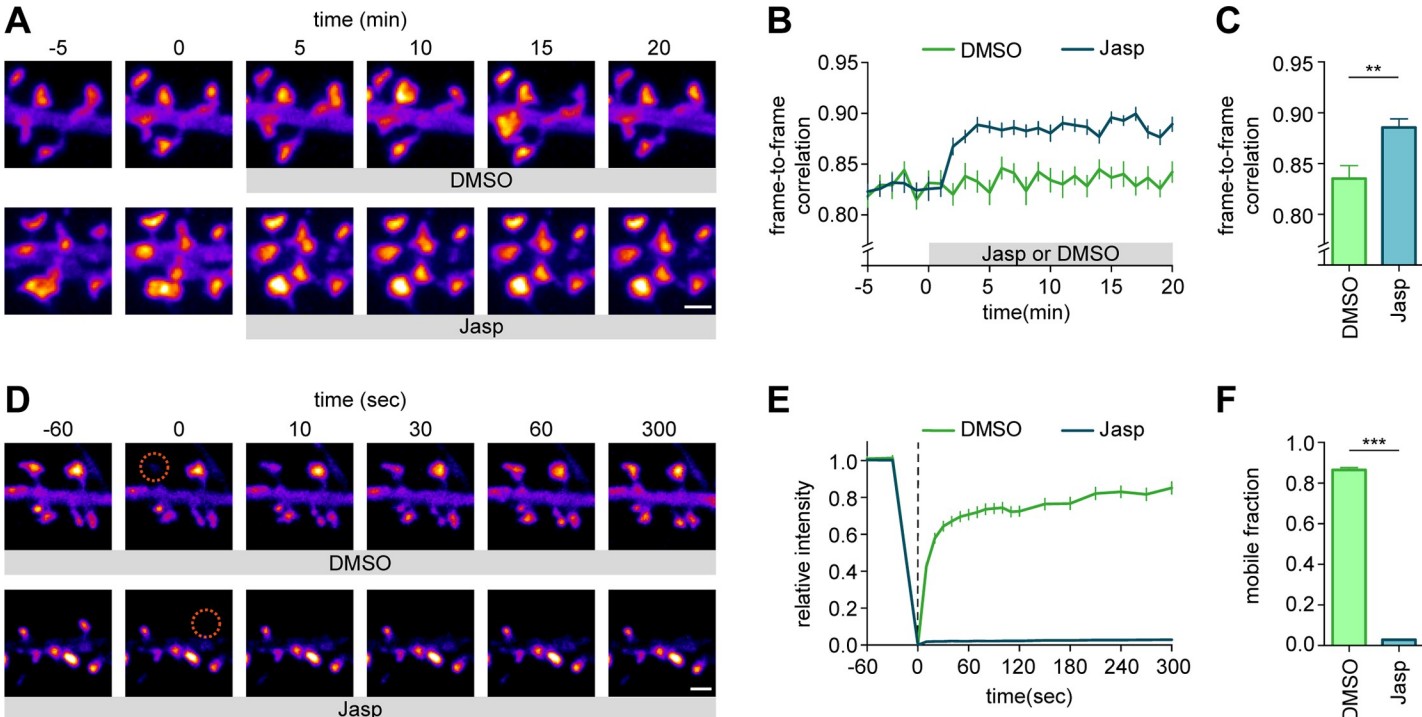

**Fig 5. Live-cell imaging of intracellular endogenous protein dynamics.** (A) Representative images of dendrites transfected with GFP-β-actin knock-in imaged over time. DMSO or Jasp was added at time point 0. DIV 21. Scale bar, 1 μm. (B) Frame-to-frame correlation of pixel intensity over time for DMSO (green) or Jasp (blue) addition. (C) Quantification of mean frame-to-frame correlation averaged over the last five time points. (D) Representative images of FRAP experiment on dendrites transfected with GFP-β-actin knock-in vector. ROIs were bleached at time point 0 (orange circle). Recovery was followed over time. DIV 21. Scale bar, 1 μm. (E) FRAP analysis of GFP-β-actin knock-in neurons treated with DMSO (control) or Jasp. ROIs were bleached at time point 0 (dotted line). (F) Quantification of mobile fraction calculated from the last five time points of each bleached ROI averaged per neuron. Data are represented as means ± SEM. $^{**}P < 0.01$, $^{***}P < 0.001$, Student $t$ test. Underlying data can be found in S1 Data. DIV day in vitro; FRAP, fluorescence recovery after photobleaching; GFP, green fluorescent protein; Jasp, jasplakinolide; ORANGE, Open Resource for the Application of Neuronal Genome Editing; ROI, region of interest.

antibody staining in resolving subsynaptic protein organization using STED microscopy, we compared the localization of the PSD95-GFP knock-in signal with that of immunolabeled PSD95. Both confocal and gSTED images of individual synapses revealed a high degree of colocalization between the PSD95-GFP knock-in and anti-PSD95 staining (Fig 6F). Additionally, gSTED revealed that even at the subsynaptic level, the PSD95-GFP knock-in and anti-PSD95 staining colocalized (Fig 6G and 6H). In contrast, two-color gSTED of GFP-β-actin knock-in and anti-PSD95 staining revealed that β-actin is enriched in dendritic spines but is largely excluded from the synapse (Fig 6I–6K). The differential distribution of the PSD95-GFP and GFP-β-actin knock-ins was confirmed by quantifying the degree of colocalization with immunolabeled anti-PSD95 using two independent metrics: the Pearson's correlation coefficient (PCC) and Manders' overlap correlation (MOC) [44,45], highlighting the need for super-resolution techniques, such as STED (S10D and S10F Fig). Additionally, we found that CaMKIIα is enriched in dendritic spines and only partially overlapped with PSD95 (S10G–S10J Fig).

Lastly, we studied the subcellular expression of proteins that have only recently been discovered. High-throughput proteomics studies have identified a number of novel components of the α-amino-3-hydroxy-5-methyl-4-isoxazolepropionic acid (AMPA) receptor complex that have different topologies and functions [46,47]. Information on the subcellular distribution of these components is sparse and largely based on overexpression, which is known to alter the

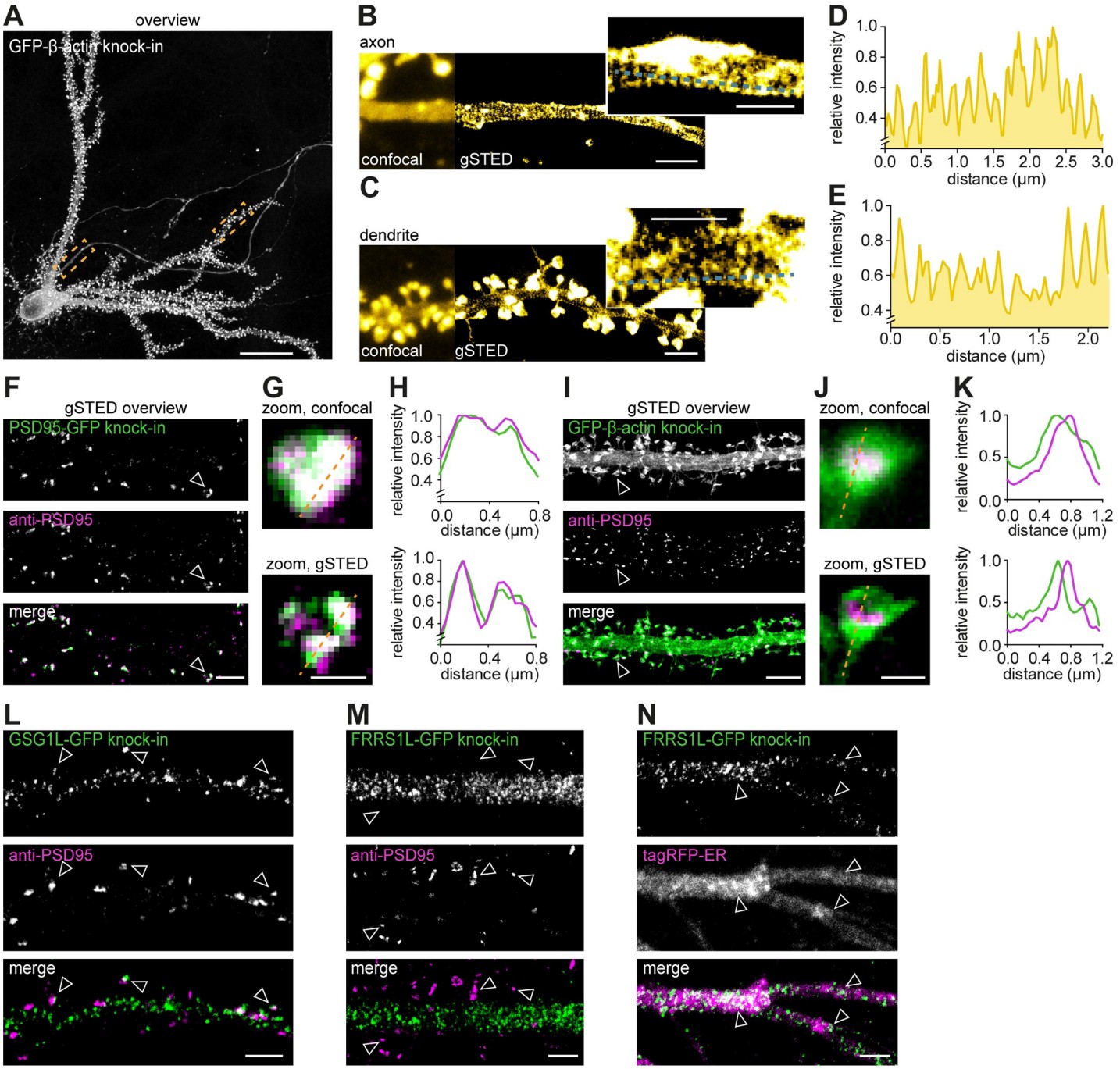

**Fig 6. STED microscopy to resolve the subcellular distribution of endogenous proteins in individual neurons.** (A) Representative gSTED image of a GFP-β-actin knock-in neuron (DIV 21) enhanced with anti-GFP (ATTO647N). Scale bar, 20 μm. (B and C) Zooms of axon (B) and dendrite (C) as indicated with boxes in (A) comparing confocal and gSTED imaging. Scale bar, 2 μm; insert scale is 1 μm. (D and E) Line scans from zooms in (B) and (C), respectively. (F) Representative gSTED images of dendrites positive for PSD95-GFP knock-in stained with anti-GFP (green, ATTO647N) and anti-PSD95 staining (magenta, Alexa594). DIV 21. Scale bar, 2 μm. (G) Zooms from (F) of individual synapses resolved with confocal and gSTED. Scale bar, 500 nm. (H) Line scans of confocal and gSTED images shown in (G). (I) Representative gSTED images of dendrites positive for GFP-β-actin knock-in stained with anti-GFP (green, ATTO647N) and anti-PSD95 (magenta, Alexa594). DIV 21. Scale bar, 2 μm. (J) Zooms from (I) of individual spines resolved with confocal and gSTED. Scale bar, 500 nm. (K) Line scans of confocal and gSTED images shown in (J). (L and M) Representative gSTED images of dendrites positive for GSG1L-GFP (L) or FRRS1L-GFP knock-in (M) stained with anti-GFP (green, Alexa488) and anti-PSD95 (magenta, ATTO647N). DIV 21. Scale bar, 5 μm. (N) Representative images of dendrites positive for FRRS1L-GFP knock-in enhanced with anti-GFP (gSTED, green) and coexpressed with tagRFP-ER (confocal, magenta). DIV 21. Scale bar, 2 μm. DIV, day in vitro; ER, endoplasmic reticulum; FRRS1L, Ferric-chelate reductase 1-like protein; GFP, green fluorescent protein; GSG1L, Germ cell-specific gene 1-like protein; gSTED, gated STED; PSD95, postsynaptic protein 95; RFP, red fluorescent protein; STED, stimulated-emission depletion.

trafficking and function of AMPA receptors at synapses. Here, we developed knock-in constructs for two AMPA receptor interactors: Germ cell-specific gene 1-like protein (GSG1L) and Ferric-chelate reductase 1-like protein (FRRS1L)/C9orf4. GSG1L has been recently shown to modulate AMPA receptor function [48,49]. Using gSTED imaging, we found that GSG1L localizes throughout the dendritic shaft and in dendritic spines, where it closely associates with synaptic PSD95 (Fig 6L). In contrast, FRRS1L was found to be excluded from synapses (Fig 6M) but showed a punctate distribution in the dendritic shaft, closely associated with the endoplasmic reticulum (ER) (Fig 6N). This is in line with a recent study showing that FRRS1L regulates AMPA receptor trafficking from the ER to control AMPA receptor surface expression [50–53]. Altogether, these results demonstrate the potential of ORANGE to uncover the nanoscale organization of endogenous proteins, in particular those with unknown distribution due to lack of specific antibodies, in individually labeled neurons.

## Dissection of endogenous NMDA receptor distribution and dynamics within individual synapses

Based on overexpression and antibody-labeling studies, the spatial organization of NMDA receptors at excitatory synapses has been proposed to be heterogenous, with receptors accumulating in distinct subsynaptic nanodomains [54–56]. However, because overexpression of individual receptor subunits could affect subunit stoichiometry and function of endogenous receptors [57], we combined ORANGE with superresolution techniques to dissect the distribution and dynamics of NMDA receptors. To visualize the total pool of NMDA receptors, we developed a knock-in construct to endogenously tag the obligatory GluN1 subunit with GFP (Fig 7A). Several studies have consistently estimated that the number of NMDA receptors at individual synapses is relatively low, ranging from 10 to 20 receptor complexes per synapse [32,58]. Despite these low copy numbers, we could detect concentrated dendritic clusters of GFP-GluN1, most of which colocalized with immunolabeled PSD95 (Fig 7A). Interestingly, we found that GFP-GluN1 intensity did not correlate with anti-PSD95 immunolabeling intensity (Fig 7B) (Pearson r: 0.19, $R^2$: 0.038, $n$ = 450 GluN1 clusters from nine neurons), consistent with earlier studies showing that the total number of NMDA receptors is largely invariable and does not scale with synapse size [59–61]. Using gSTED imaging, we found that although most GFP-GluN1 clusters localized to synapses, some smaller extrasynaptic clusters could be detected (Fig 7C–7E). Next, we measured the total GFP-GluN1 cluster area in individual synapses and found a slight correlation with synapse size (Pearson r: 0.64, $R^2$: 0.4087, $n$ = 266 synapses from three neurons; Fig 7F). Thus, our data suggest that the subsynaptic area covered by NMDA receptors, but not the total number of receptors, scales with synapse size. gSTED imaging of individual synapses also indicated that the subsynaptic distribution of GFP-GluN1 is heterogeneous (Fig 7B and 7G), with individual synapses containing one or more smaller GFP-GluN1 substructures (Fig 7H) ($n$ = 266 synapses from three neurons).

To further investigate the subsynaptic distribution of NMDA receptors, we turned to single-molecule localization microscopy (SMLM). The GFP-GluN1 knock-in was immunolabeled with anti-GFP and Alexa647-coupled secondary antibodies for direct stochastic optical reconstruction microscopy (dSTORM) to reconstruct the spatial organization of NMDA receptors at individual synapses with nanometer precision (Fig 7I and 7J). Clusters of GFP-GluN1 receptors were identified using density-based spatial clustering of applications with noise (DBSCAN) [62]. Next, all localizations within individual clusters were plotted and color-coded for the local density. These local density maps revealed that, within individual clusters, NMDA receptors form distinct nanodomains (Fig 7K), consistent with our gSTED data (Fig 7D). We found that the majority of GFP-GluN1 clusters contained one to three nanodomains with a

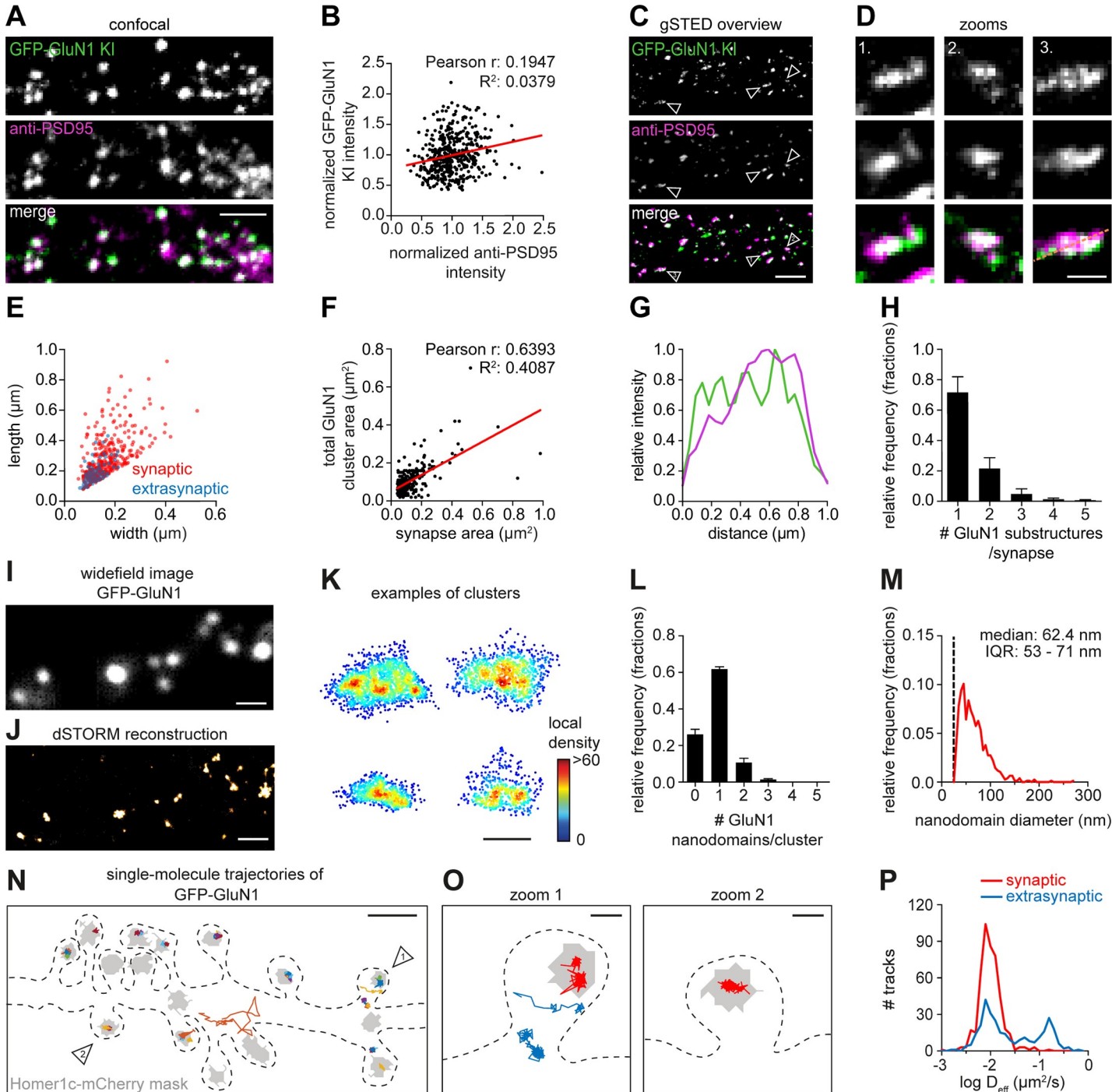

**Fig 7. NMDA receptors concentrate in subsynaptic nanodomains and are highly immobilized in synapses.** (A) Representative images of a dendrite positive for GFP-GluN1 KI (green) stained for PSD95 (magenta, Alexa647). Scale bar, 2 μm. (B) Correlation between GFP-GluN1 KI and anti-PSD95 staining intensity within individual GFP-GluN1 puncta. (C) Representative gSTED images of dendrites positive for GFP-GluN1 KI enhanced with anti-GFP (green, ATTO647N) and anti-PSD95 (magenta, Alexa594). DIV 21. Scale bar, 2 μm. (D) Zooms of individual synapses indicated in (C). Scale bar, 500 nm. (E) FWHM analysis of GFP-GluN1 structures comparing width and length of individual synaptic (red) and extrasynaptic (blue) GluN1 clusters. (F) Correlation between GFP-GluN1 cluster area and synapse area (based on anti-PSD95 staining) for individual synapses. (G) Line scan of synapse zoom 3 in (D). (H) Quantification of the number of GFP-GluN1 substructures per synapse. (I) Representative image of dendrite positive for GFP-GluN1 KI stained with anti-GFP (Alexa647). DIV 21. Scale bar, 1 μm. (J) Single-molecule dSTORM reconstruction of example shown in (I). Scale bar, 1 μm. (K) Examples of individual GFP-GluN1 clusters with single localizations plotted and color-coded based on the local density. Scale bar, 200 nm. (L) Quantification of number of GFP-GluN1 nanodomains per cluster. (M) Frequency distribution of GFP-GluN1 nanodomain size. Dotted line indicates nanodomain size cutoff. Bin size: 5 nm. (N) Representative example of GFP-GluN1 (anti-GFP nanobody conjugated to ATTO647N) single-molecule trajectories in a dendrite plotted with a random color and on top of a synapse mask (gray) based on Homer1c-mCherry widefield image.

Dotted line indicates cell outline. DIV 21. Scale bar, 1 μm. (O) Zooms of individual spines indicated in (N) with example trajectories of synaptic (red) or extrasynaptic (blue) receptors. Scale bar, 200 nm. (P) Frequency distribution showing the diffusion coefficient of synaptic and extrasynaptic tracks. Data in bar plots are presented as means ± SEM. Underlying data can be found in S1 Data. DIV, day in vitro; dSTORM, direct stochastic optical reconstruction microscopy; FWHM, Full Width at Half Maximum; GFP, green fluorescent protein; GluN, glutamate receptor NMDA; gSTED, gated stimulated-emission depletion; KI, knock-in; NMDA, *N*-methyl-D-aspartate; PSD95, postsynaptic protein 95.

median size of approximately 62 nm (IQR: 53–71 nm) ($n$ = 859 GFP-GluN1 clusters from three neurons) (Fig 7L and 7M). Thus, these SMLM data indicate that endogenous NMDA receptors form distinct subsynaptic nanodomains.

To gain insight in the subsynaptic mobility of endogenously expressed NMDA receptors, we probed the diffusion kinetics of individual receptors using universal point accumulation in nanoscale topography (uPAINT) [63]. Stochastic labeling of individual GFP-tagged receptors with a GFP nanobody coupled to ATTO647N provided a map of individual receptor mobility along stretches of dendrites (Fig 7N and 7O). Most receptor trajectories mapped within the boundaries of the synapse. Strikingly, we found that these synaptic NMDA receptors were largely immobilized (median diffusion coefficient synaptic tracks: 0.0096 μm$^2$/s, IQR: 0.0079–0.0122, $n$ = 462 tracks from 6 neurons), whereas on average, extrasynaptic receptors diffuse at higher rates (0.0224 μm$^2$/s, IQR: 0.0123–0.0419, $n$ = 307 tracks from 6 neurons) (Fig 7P). Altogether, by combining the ORANGE toolbox with superresolution microscopy, we show that NMDA receptors are enriched in the PSD, where they are highly immobilized and cluster in subsynaptic nanodomains.

## Cre-dependent coexpression for multiplex labeling of two proteins in single neurons

We have shown that ORANGE mediates the integration of small epitope tags and fluorescent proteins in single genes (Fig 1). Tagging two proteins simultaneously in one neuron for dual-color imaging, however, is challenging using this approach. NHEJ-mediated integration of the donor sequence is homology independent, and therefore, the integration of independent donor sequences cannot be targeted to specific genes but occurs at random [23]. Recently, NHEJ-based, targeted integration of Cre recombinase was used to disrupt the target gene and drive the expression of a second protein used as a reporter of a successful knock-out [25]. Based on this, we reasoned that genomic integration of a fluorescent protein together with Cre recombinase could be used to trigger the expression of a second gRNA from an additional knock-in plasmid. This approach would facilitate the sequential integration of two donor sequences targeted to two genomic loci in a single neuron. To test this, we first developed knock-in constructs integrating a C-terminal GFP tag fused to a P2A-Cre sequence (GFP-P2A-Cre), leading to bicistronic expression of a GFP-fusion protein and Cre recombinase (Fig 8A). This yielded robust recombination and expression of flip-excision (FLEx) mCherry and Synapsin-FLAG (Fig 8B). We did, however, observe some cells that only expressed the FLEx construct without visible GFP signal, suggesting that either Cre expression is somewhat leaky or that very low levels of Cre are already sufficient to recombine FLEx switches.

Building on GFP-P2A-Cre knock-ins, we developed a pORANGE vector containing a Cre-dependent Lox-STOP-Lox sequence in the U6 promoter [64], which blocks expression of the gRNA until Cre is expressed (Fig 8C). When combined with a GFP-P2A-Cre knock-in, this would mediate reliable dual-color knock-ins with NHEJ because the Lox-STOP-Lox gRNA is only expressed after GFP-P2A-Cre integration is completed and a functional protein has been produced from this allele (S11A Fig). Thus, this mechanism should prevent mix-up of donor

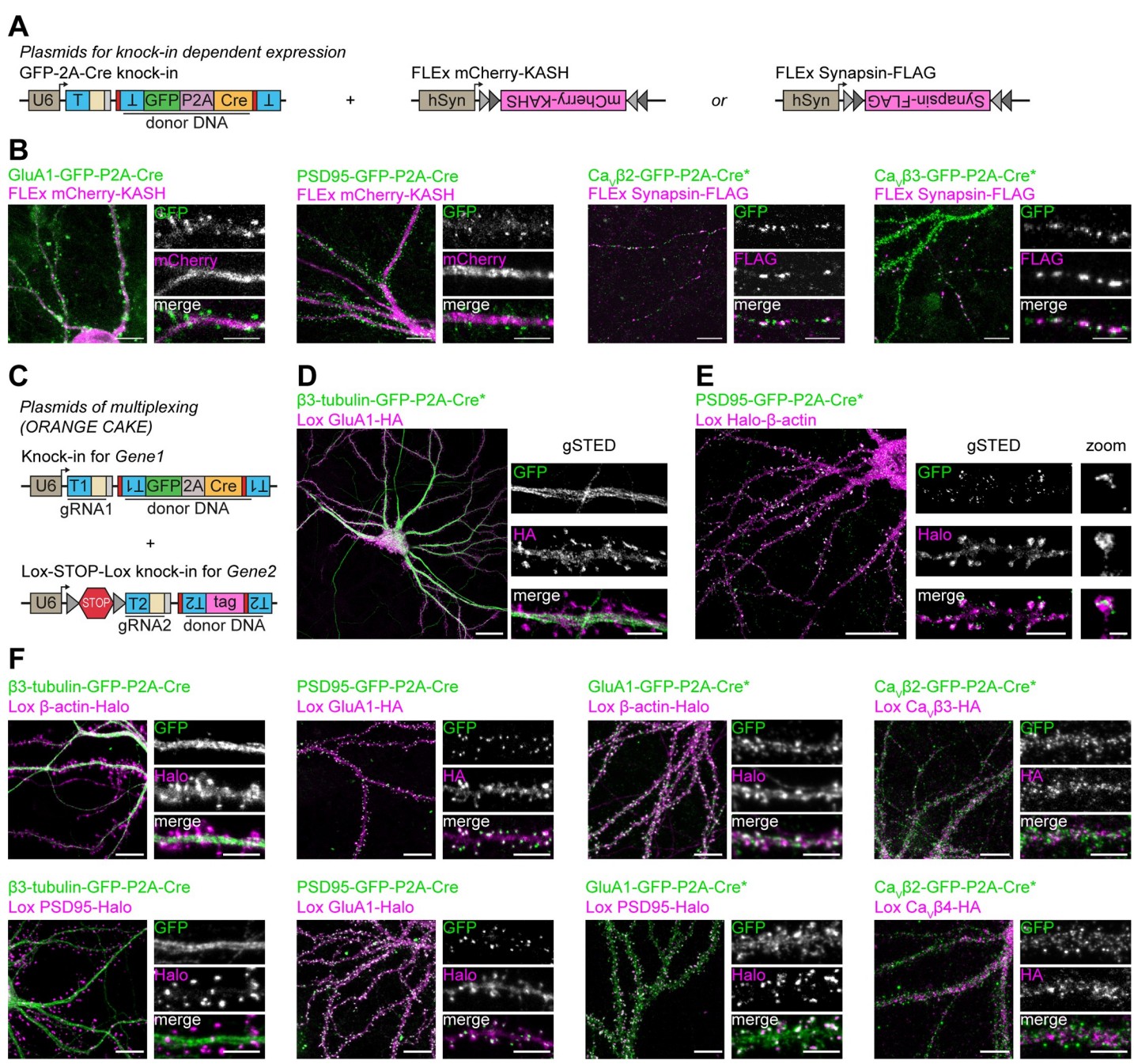

**Fig 8. Cre-dependent coexpression and labeling of two proteins in single neurons.** (A) Overview of plasmids used for Cre-dependent expression of mCherry-KASH or Synapsin-FLAG in knock-in neurons. (B) Examples of GFP-P2A knock-in–driven expression of mCherry-KASH or Synapsin-FLAG (Alexa568) (magenta) for various knock-ins. DIV 21. Scale bars, 10 μm and 5 μm for the overviews and zooms, respectively. (C) Overview of plasmids used for multiplex knock-in of two proteins in single neurons (ORANGE-CAKE). (D and E) Examples of β3-tubulin-GFP-P2A-Cre (green), Lox GluA1-HA (magenta, Alexa594) double knock-in, (D) and PSD95-GFP-P2A-Cre (green), Lox Halo-β-actin (magenta, JF549) double knock-in (E). Shown are overviews (confocal) and zooms (gSTED). DIV 21. Scale bars, 20 μm for the overviews and 5 μm (dendrites) and 500 nm (spine) for the zooms. (F) Examples of various combinations of GFP-P2A-Cre (green) and Lox (magenta) double knock-ins. HA was visualized by anti-HA staining (Alexa594), and Halo with Halo-JF549 ligand. DIV21. Scale bars, 10 μm and 5 μm for the overviews and zooms, respectively. Asterisk indicates enhancement with anti-GFP antibody (Alexa488). CAKE, conditional activation of knock-in expression; Ca$_V$, voltage-dependent Ca$^{2+}$-channel; DIV, day in vitro; FLEx, flip-excision; GFP, green fluorescent protein; GluA, glutamate receptor AMPA; gRNA, guide RNA; gSTED, gated stimulated-emission depletion; HA, hemagglutinin; hSyn, human Synapsin; JF549, Janelia Fluor 549; KASH, Klarsicht, ANC-1, Syne Homology; ORANGE, Open Resource for the Application of Neuronal Genome Editing; PSD95, postsynaptic protein 95; T, target sequence.

sequences in the targeted loci. We dubbed this method conditional activation of knock-in expression (CAKE). Using ORANGE-CAKE, we developed multiplex knock-ins for a range of combinations and used these for dual-color confocal and gSTED microscopy of endogenous proteins (Fig 8D and 8F). As was observed for FLEx switches, we also observed occasional expression of Lox-STOP-Lox knock-ins without observable GFP signal (see Discussion). Additionally, for β3-Tubulin-GFP-P2A-Cre and GluA1-GFP-P2A-Cre, we occasionally observed erroneous integration of donor DNA in the nontargeted locus (e.g., HaloTag labeling from the gene targeted with GFP-P2A-Cre) (S11B and S11C Fig, see Discussion). Importantly, we never observed expression of Lox-STOP-Lox knock-ins in cultures without expression of Cre recombinase. Together, these data show the feasibility of multiplex labeling in single cells using ORANGE-CAKE to study spatiotemporal protein expression of multiple proteins simultaneously in individual neurons.

## Discussion

Mapping the subcellular distribution of proteins at high spatial resolution is fundamental to understand cell biological processes. Ongoing developments in superresolution imaging technologies have dramatically improved the spatial resolution, allowing the dissection of molecular organization of subcellular structures at nanometer precision. However, a major obstacle remains the availability of a flexible strategy to efficiently and specifically label endogenous proteins, especially in neurons. Here, we developed ORANGE, a simple and scalable toolbox for epitope tagging of endogenous proteins using CRISPR/Cas9, and we provide a readily usable knock-in library that enables in-depth interrogation of protein distribution and dynamics in postmitotic neurons at high spatial resolution. Although CRISPR/Cas9-based tagging approaches have been developed for neurons, until now, large-scale applications of these methods have been limited. ORANGE offers a single template vector that only requires standard cloning methods. Moreover, we demonstrated that this approach is compatible with various generally used DNA delivery methods, including lipofection, electroporation, LVs, and AAVs, and thus can be used in dissociated neuronal cultures and organotypic slice cultures and in vivo. Instead of relying on antibodies that target individual proteins with varying levels of specificity and efficiency, the ORANGE toolbox utilizes fluorescent proteins that directly report protein localization, self-labeling enzymes, or epitope tags for which universal antibodies are available. Moreover, integration of Cre recombinase allowed for tagging of two endogenous proteins in single cells.

We demonstrated the level of accuracy of targeted genomic integration using ORANGE in several ways. First, we analyzed whether insertion of GFP was correct at the genomic level using next-generation sequencing. We detected high frequency of correct integration for many of the knock-ins, although the occurrence of indels is highly variable between individual targets. However, indels should not form a limitation for many purposes, including fluorescent imaging, because only neurons with detectable fluorescent signal are selected, and proteins with out-of-frame integration remain undetectable. At the network level, we expect that the effect of frameshift mutations is limited, especially when using lipofection, because more than 90% of cultured cells are not transfected and remain unedited. Importantly, we noted that the frequency of correct integration did not correlate with Doench on-target score [30] and Bae out-of-frame score [31], scores often used to select a target sequence with a high efficiency. Also, when testing knock-in constructs targeting different target sequences in the same gene, we found significant differences in knock-in efficiency, independent of the Doench and Bae scores. These scores are primarily developed based on knock-out outcomes, which might

explain why these scores are not correlated with the accuracy or efficiency of donor integration.

Second, for all our targets, we found that the distribution of the GFP signal was consistent with previous reports of protein localization inferred from immunolabeling or biochemical fractionation experiments. Our results indicate that, when expressed, the tag accurately reports protein localization and does not affect protein levels in most knock-in-positive neurons. These results show that well-designed knock-ins do not affect localization of the targeted protein and that off-target expression of the donor tag is extremely rare. Multiple mechanisms within the design of knock-in construct prevent off-target expression. We selected target sequences with a high MIT score, meaning that the sequence is unique within the genome and that potential off-targets are intergenic or in introns. If off-target integration in protein-coding sequences does occur, the donor orientation will be random (i.e., 50% is in the inverted orientation). Additionally, in 66% of off-target integrations, the donor would be out of frame, and donor integration in a random location within a protein is likely to severely affect folding, leading to degradation of the targeted protein.

Third, with immunocytochemistry, we found that knock-ins were most often expressed at endogenous levels. However, in a few cases, we did observe that the tagged protein was expressed at slightly lower levels compared with the untagged protein in untransfected neurons. This might indicate that, in these neurons, one of the two alleles contains indels after genome editing and/or failed to integrate the donor DNA, consistent with estimates with the HITI method that 30%–50% of knock-in-positive cells show biallelic integration [24]. We also showed that, for C-terminal tagging of PSD95, Shank2, and N-terminal Bassoon (but not C-terminal Bassoon knock-ins), knock-in-negative neurons are likely partial or complete knock-outs. This difference in protein levels, especially for C-terminal-tagged proteins, might be the result of different sensitivity to, for example, nonsense-mediated decay [65]. Ongoing advancements in CRISPR/Cas9 technology are likely to lead to new developments that increase the on-target integration efficiency and precision of this approach. For instance, Cas9 variants with higher specificity could decrease indel frequency [66,67], and the knock-in efficiency and repair accuracy may be predicted based on the target sequence [28,29]. Also, alternative delivery methods such as ribonucleoproteins (RNPs) [68] might increase the efficiency of DNA delivery.

An important advantage of our method is that targeted integration of common epitope tags circumvents the need for developing new specific antibodies. In particular, for proteins that are highly homologous in their amino acid sequence and for which generating specific antibodies is challenging, it is now possible to develop specific knock-in constructs that will report subcellular localization at unmatched specificity. As an example, we demonstrated successful knock-ins for RIM1 and RIM2, two highly homologous active zone proteins for which isoform-specific antibodies are not available. The knock-in constructs presented in our library are designed using the rat genome as a template. However, because of high gene homology, multiple of the knock-in constructs are compatible with the mouse genome (see S2 Table). For example, we have shown that our GluA1 knock-in works both in dissociated rat hippocampal cultures as well as in mouse organotypic hippocampal slice cultures and in vivo in mouse brain.

ORANGE is easily employed on targets yet to be characterized. Next-generation sequencing efforts and high-resolution proteomics studies continue to discover the implication of novel proteins in biological processes, but for many of these proteins, specific and efficient antibodies are lacking. For instance, we developed knock-in constructs for two AMPA receptor complex constituents, FRRS1L/C9orf4 and GSG1L, that have only recently been discovered in a high-resolution proteomics study [47]. For both proteins, functional characterization is

available [48–50,52,53], but high-resolution information on subcellular distribution was lacking because of the unavailability of specific antibodies. Thus, ORANGE allowed us to visualize and image these proteins at high resolution, showing that, whereas GSG1L is localized on the dendritic shaft and in dendritic spines, FRRS1L is preferentially targeted to the soma and dendritic shaft, seemingly associated with the ER.

The ability to tag endogenous proteins in sparse subsets of cells is particularly advantageous for superresolution approaches. Also, sparse labeling of cells increases contrast and provides internal negative controls because neighboring, nontargeted cells are unlabeled. The resolution of these approaches will detect any distortion in molecular organization due to, for instance, overexpression artifacts, and therefore, these methods are highly sensitive to nonspecific labeling. We exploited the advantages of ORANGE to dissect the subcellular distribution of a number of neuronal proteins using different superresolution imaging approaches. We mapped the distribution of endogenous cytoskeletal elements, signaling proteins, and synaptic receptors. Our experiments demonstrate that endogenous CaMKIIα has two distinct kinetic populations. Focusing on glutamate receptors, we found that endogenous NMDA receptors are highly immobilized at synaptic sites and enriched in distinct subsynaptic nanodomains. This particular distribution is likely to shape the efficiency of receptor activation by glutamate [1], and therefore, dissection of the underlying molecular mechanisms is essential for our understanding of synapse physiology. Thus, ORANGE enables superresolution imaging and live-cell single-molecule tracking of neuronal proteins and thus provides a scalable approach to efficiently and reliably map the dynamic distribution of endogenous proteins at nanometer resolution.

Finally, we show that ORANGE can be used for multiplex labeling and dual-color imaging of endogenous proteins. Multiplex gene editing has remained a challenge in neuronal cells, and existing methods have relatively low efficacy [18] or are limited to specific combinations with small epitope tags [23]. Our CAKE method of sequential genome editing using a GFP-P2A-Cre knock-in and a second Cre-dependent knock-in mediates flexible, multiplex editing for a wide range of combinations, without restrictions on donor DNA sequence. We did observe, however, that some GFP-P2A-Cre knock-ins had reduced GFP fluorescence compared with regular GFP knock-ins. Although it is currently unclear what the cause of this is, it is likely that the substantial increase in mRNA length reduces protein levels [69]. Therefore, the expression level of each knock-in should be carefully assessed for each target. For some GFP-P2A-Cre knock-ins, including β3-tubulin and GluA1, we did observe occasional erroneous integration of the second, Cre-dependent knock-in. This is likely induced by rapid expression of Cre recombinase from these knock-ins after integration in the first allele, leading to activation of the Cre-dependent knock-in before the second allele has been edited. Indeed, we did already observe GFP expression from β3-tubulin knock-ins after 24 hours, and it is not unlikely that this time span is insufficient to edit both alleles. Despite these current limitations, we feel that CAKE is a valuable tool to study the localization of multiple endogenous proteins in individual cells.

We believe that ORANGE is a simple and efficient genome editing toolbox that will rapidly advance many fields in biology through the in-depth investigation of protein distribution in cultured cell lines, primary cells, organotypic slice cultures, and animal models, but in particular, ORANGE presents one of the few possibilities to tag proteins in neurons. Further development of tools for cell type–specific targeting of epitope tags would allow interrogation of protein distribution in specialized neuron types in the brain. Apart from epitope tagging, our toolbox can, for example, be used for insertion of enzymes for proximity biotinylation [70], labeling of organelles for electron microscopy [71], or light-sensitive dimerization sequences for optical control over protein or organelle positioning [72,73]. The unprecedented number

of applications of ORANGE will undoubtedly deepen our molecular understanding of how the spatial distribution of endogenous proteins contributes to cell biological processes.

# Materials and methods

## Ethics statement

All experiments were approved by the Dutch Animal Experiments Committee (Dier Experimenten Commissie [DEC] AVD1080020173404, AVD1080020173847, and AVD11500 2016797), performed in line with institutional guidelines of Utrecht University, and conducted in agreement with Dutch law (Wet op de Dierproeven, 1996) and European regulations (Directive 2010/63/EU). Timed pregnant Wistar rats were obtained from Janvier Labs. Wild-type male and female mice were used. Rosa26-Cas9 knock-in mice are originally from [27].

## Antibodies and reagents

Primary antibodies used in this study are the following: rabbit anti-GFP (MBL Sanbio, 598, RRID AB_591819), rat anti-HA ([3F10], Sigma, 11867423001, RRID AB_390919), mouse anti-FLAG ([M2], Sigma, F3165, RRID AB_259529), mouse anti-PSD95 ([K28/43], Neuromab, 75–028, RRID AB_2307331), mouse anti-alpha-tubulin ([B-5-1-2], Sigma, T5168, RRID AB_477582), mouse anti-Bassoon ([SAP7F407], Enzo, ADI-VAM-PS003-F, RRID AB_10618753), mouse anti-Shank2 ([N23B/6], Neuromab, 75–088, RRID AB_2254586), mouse anti-CaMKIIα ([6G9], Sigma, C265, RRID AB_2314080), and ATTO647N-conjugated anti-GFP nanobodies (GFPBooster-ATTO647N, Chromotek). Alexa488-, Alexa568-, Alexa594-, and Alexa647-conjugated secondary antibodies were from Life Technologies. ATTO647N-conjugated secondary antibodies were from Sigma. Alexa594- and Alexa647-conjugated phalloidin was from Life Technologies. Halo-ligands conjugated to Janelia fluorophore 549 (Halo-JF549) and 646 (Halo-JF646) were from Promega.

## Dissociated neuronal cultures

Dissociated hippocampal cultures were prepared from embryonic day 18 (E18) rat brains of both genders, as described in [74]. Dissociated neurons were plated on Ø18-mm coverslips coated with poly-L-lysine (37.5 µg/ml, Sigma-Aldrich) and laminin (1.25 µg/ml, Roche Diagnostics) at a density of 100,000 neurons per well. Neurons were grown in Neurobasal medium (NB) supplemented with 1% penicillin and streptomycin (pen/strep), 2% B27, and 0.5 mM L-glutamine (all from Gibco) (NB-complete medium) at 37˚C in 5% $CO_2$. From DIV 1 onward, medium was refreshed weekly by replacing half of the medium with Brainphys neuronal medium supplemented with 2% NeuroCult SM1 neuronal supplement (STEMCELL Technologies) and 1% pen/strep (BP-complete medium).

## Organotypic hippocampal slice cultures

Organotypic hippocampal slice cultures were prepared from wild-type mice at postnatal day 6–8. After decapitation, the brain was quickly removed and placed in ice-cold Gey's Balanced Salt Solution (GBSS) containing (mM) 137 NaCl, 5 KCl, 1.5 $CaCl_2$, 1 $MgCl_2$, 0.3 $MgSO_4$, 0.2 $KH_2PO_4$, and 0.85 $Na_2HPO_4$ and supplemented with 12.5 mM HEPES, 25 mM glucose, and 1 mM kynurenic acid (pH set at 7.2, osmolarity set at 320 mOsm, sterile filtered). The frontal part of the brain and the cerebellum were removed along the transverse plane, and the hemispheres were then separated along the midline. Hippocampi were dissected and sliced perpendicularly to the long axis of the hippocampus with a thickness of 400 µm using a McIlwain Tissue Chopper. Slices were washed in culturing medium (consisting of 48% MEM, 25%

HBSS, 25% horse serum, 30 mM glucose, and 12.5 mM HEPES, with pH set at 7.3–7.4 and osmolarity set at 325 mOsm) before being placed on Millicell cell culture inserts (Millipore) in 6-well plates containing culturing medium. Slices were kept at 37°C with 5% $CO_2$ until use, and culturing medium was completely replaced twice per week.

## Design and generation of ORANGE knock-in plasmids

**Cloning of CRISPR/Cas9 knock-in vector pORANGE.** To facilitate the generation of knock-in constructs, we developed a simple template vector (pORANGE). For this, we used pSpCas9(BB)-2A-Puro (PX459) V2.0 (Addgene 62988) and replaced SpCas9puro by SpCas9 from pAAV-nEFCas9 (Addgene 87115) flanked by the bipartite SV40 nuclear localization signal (NLS) sequences using the AgeI and EcoRI restriction sites, generating pSpCas9. To facilitate cloning of donor sequences, a multiple cloning site was inserted by annealing two complementary DNA oligos and ligation into the XbaI site of pSpCas9 generating pORANGE.

**Design and cloning of ORANGE knock-in constructs.** To select regions within a protein of interest suitable for introducing a tag, we carefully examined known protein functions, domains, presence of signal peptides, binding ligands, and (if known) protein structure to minimize potential effects of the inserted tag sequence on protein function. For an overview of literature and design rationality given for each knock-in construct, see S1 Table. For most proteins, this resulted in tagging close to the start or stop codon or just behind the signal peptide. In some cases (including CaMKIIα, Rab11, and β-actin knock-in #2), the genes were tagged just before the start codon. PAM sites in these identified regions were located in genomic sequences downloaded from the RGSC5.0/rn5 genome assembly through the UCSC genome browser gateway (https://genome-euro.ucsc.edu/). Target sequences were chosen, taking into consideration the MIT guide specificity score [75]. For some of the knock-ins, an extra G nucleotide was incorporated at the start of the target sequence to enhance transcription from the U6 promotor. We have no indication that this altered knock-in efficiency (for all protein target sites, target sequences, and gRNA scores, see S2 Table).

Next, oligos containing the 20-bp target sequences were annealed and ligated into the BbsI sites of pORANGE (Figs 1 and S1). Donor sequences were designed to contain the fluorescent tag sequence (GFP or mEos3.2) flanked by two Cas9 target sites identical to the genomic target site. Importantly, to facilitate genomic integration of the donor sequence in the correct orientation, these target sites including PAM sequences were inserted as the reverse complement of the genomic target sequence (Figs 1A and S1). Additional linker sequences of at least three amino acids and additional base pairs to make the donor in frame after integration in the genome were introduced between the target sites and the tag sequence. Also, a start codon and new Kozak sequence or stop codon was introduced in the linker when proteins were tagged before the genomic start or stop codon, respectively. For the CaMKIIa knock-in construct, the reverse integration of the incomplete target sequence introduces an additional start codon. Extra base pairs were introduced in the linker to make this extra start codon in frame with the donor. To facilitate exchange of donor tags, in-frame BmtI and AfeI restriction sites were introduced in the linker for some, but not all, knock-in constructs. Primer oligos with overhangs containing all these features were designed to generate the complete donor sequence by PCR. (See S1 Fig for two example designs.) The donor sequences were PCR amplified from a GFP-containing plasmid as template and ligated into the multiple cloning site of the pORANGE vector containing the inserted target sequence to generate the complete knock-in construct. For all primers used to generate the knock-in donor inserts, see S3 Table. To replace GFP in the donor DNA, pORANGE plasmids were digested with BmtI and AfeI, and

replacements were generated by primer ligation (in case of 2× HA or 2× FLAG) or PCR for larger donors.

For LV applications, the ORANGE system was split into two plasmids. To generate pFUGW-Cas9, SpCas9 (from pAAV-nEFCas9) was ligated into the AgeI and EcoRI sites of pFUGW (Addgene 14883). To generate the gRNA and donor containing LV plasmid, first, mCherry-KASH amplified from pAAV-mTubb3 (Addgene 87116) was ligated into the BshTI and EcoRI sites of pFUGW-Cas9 replacing Cas9, yielding pFUGW-mCherry-KASH. Then, the U6 promotor, gRNA, and the donor sequence were amplified by PCR from the pORANGE construct and inserted into the PacI site of pFUGW-mCherry-KASH using Gibson assembly (NEBuilder HiFi DNA assembly cloning kit).

For AAV vectors, we developed a pAAV backbone (pAAV-MCS-mCherry-KASH) containing a multiple cloning site, EF-1α promoter, and mCherry-KASH using Gibson assembly. Knock-in cassettes containing the U6 promoter, gRNA, and donor DNA were subcloned by digesting pORANGE with PscI/MluI, which was ligated in the NcoI and SgsI sites of pAAV MCS mCherry-KASH.

To create Cre-dependent knock-ins for CAKE, we obtained an mU6 promoter containing a STOP sequence flanked by LoxP551 sites from Addgene (#113160) [64] with PCR. pORANGE backbone was digested with PscI and BbsI to remove the original promoter, and Gibson assembly was used to ligate the PCR product to obtain pORANGE Lox. Knock-ins in pOR-ANGE Lox are cloned with identical methods as regular knock-ins in pORANGE (discussed above).

For the expression of FLEx switches, the pFSW backbone with synapsin-1 promoter (a gift from Dr. Pascal Kaeser, Harvard Medical School) was digested with KpnI and PacI. Inverted mCherry-KASH and a FLEx switch based on Addgene #50955 [76] were generated by PCR and ligated with Gibson assembly to obtain pFSW-FLEx-mCherry-KASH. To replace mCherry-KASH with Synapsin-FLAG, Synapsin-1 with FLAG tag was generated by PCR from pCMV(pr)Synapsin-1Cherry-N1lenti H81 (a gift from A. Jeromin, Allen Brain Institute, Seattle, United States), and ligated in the BmtI/BshTI restriction sites. pCaMK Homer1c-mCherry was cloned via amplification of Homer1c-mCherry from pCMV Homer1c-mCherry [55] using PCR and ligation into the XhoI and MfeI sites of pCaMK mCherry-GluA1-CIBN (Addgene #89444) [72]. All constructs were verified by sequencing.

## Transfection of dissociated hippocampal cultures

Neurons were transfected at DIV 3 (for knock-in) or DIV 14–18 (for overexpression) using Lipofectamine 2000 reagent (Invitrogen). Briefly, for one Ø18-mm coverslip covered with 100,000 neurons, 1–2 μg DNA was mixed with 3.3 μl Lipofectamine in 200 μl NB medium and incubated for 30 minutes at room temperature (RT). Next, 500 μl conditioned medium was transferred to a new culture plate and replaced by 300 μl NB supplemented with 0.5 mM L-glutamine. The DNA mix was added to the neurons and incubated at 37°C and 5% $CO_2$. After 90–120 minutes, neurons were transferred to the new culture plate with conditioned medium and 500 μl new NB medium supplemented with L-glutamine, B27, and pen/strep and kept at 37°C and 5% $CO_2$ for at least 3 days (for overexpression) and between 1–20 days for knock-in, depending on the experiment.

## Electroporation of dissociated hippocampal neurons

For electroporation, hippocampal neurons were collected directly after dissection and dissociation in a 15-ml tube and centrifuged for 5 minutes at 200$g$. Neurons were resuspended in AMAXA transfection solution (Lonza) ($3 \times 10^5$ neurons per sample), mixed with 8 μg DNA,

transferred to a gene pulser cuvette (Biorad), and electroporated using a Lonza Nucleofector 2b. Immediately after electroporation, fresh 37˚C NB medium supplemented with B27, L-glutamine, and pen/strep was added to the cuvette, after which the neurons were plated on a coated Ø18-mm coverslip using a Pasteur pipette. Neurons were incubated at 37˚C and 5% $CO_2$ for 3 hours, after which all medium was replaced with fresh NB medium supplemented with B27, L-glutamine, and pen/strep.

## HaloTag labeling of dissociated hippocampal cultures

HaloTag labeling was performed with cell-permeable Halo-JF549 or Halo-JF646 ligands. Prior to use, ligands were dissolved in DMSO to 200 μM and stored in single-use aliquots at −20˚C. HaloTag ligands were added to culture medium at a final concentration of 200 nM, and cells were placed back in the incubator for 15 minutes. After rinsing the cells with culture medium, cells were fixed using 4% (w/v) paraformaldehyde (PFA) and 4% (w/v) sucrose in phosphate-buffered saline (PBS) (PFA/Suc).

## Immunocytochemistry of dissociated hippocampal cultures

Immunocytochemistry was performed as described below, unless indicated otherwise. Hippocampal neurons were fixed using PFA/Suc for 10 minutes at RT and washed three times in PBS containing 0.1 M glycine (PBS/Gly). Neurons were blocked and permeabilized in blocking buffer (10% [v/v] normal goat serum [NGS] (Abcam) in PBS/Gly with 0.1% [v/v] Triton X100) for 1 hour at 37˚C. Next, coverslips were incubated with primary antibodies diluted in incubation buffer (5% [v/v] NGS in PBS/Gly with 0.1% [v/v] Triton X100) overnight at 4˚C. Coverslips were washed three times for 5 minutes with PBS/Gly and incubated with secondary antibodies diluted 1:400 in incubation buffer for 1 hour at RT. Coverslips were washed three times for 5 minutes in PBS/Gly, dipped in milliQ water (MQ), and mounted in Mowiol mounting medium (Sigma).

## AAV production

AAV vectors serotype 5 encoding for GluA1-Halo or PSD95-Halo knock-ins were produced as described in detail in [77] using helper plasmids obtained from [78]. In brief, HEK293T cells were plated 1 day before transfection in Dulbecco's Modified Earl's Medium (DMEM) supplemented with 10% fetal calf serum (FCS) and 1% pen/strep. At 2 hours before transfection, medium was exchanged with Iscove's Modified Dulbecco's Medium (IMDM) containing 10% FCS, 1% pen/strep, and 1% glutamine. Transfection was performed with polyethylenimine (PEI). At 1 day after transfection, medium was exchanged with fresh IMDM with supplements. At 3 days after transfection, medium was aspirated, and cells were harvested using a cell scraper. After three freeze/thaw cycles and treatment with DNAseI, AAV vectors were purified using an iodixanol density gradient and ultracentrifugation (70 minutes, 69,000 rpm at 16˚C using rotor 70Ti [Beckman Coulter]). The fraction containing AAV particles was concentrated with centrifugation (3,220*g*, 15 minutes at RT) using an Amicon Ultra 15 column (Merck Millipore). Columns were washed 3 times using D-PBS containing 5% sucrose. AAV vectors were stored at −80˚C until use. Titers were measured using qPCR.

## Stereotactic injection and staining of acute brain slices

AAV vectors were injected in 4- to 7-month-old Rosa26-Cas9 knock-in mice of either sex [27]. Mice were anaesthetized with an intraperitoneal injection of ketamine (75 mg/kg, Narketan; Vetoquinol BV) and dexmedetomidine (1 mg/kg, Dexdomitor; Orion Pharma). Analgesia

was provided before the start of surgery (carprofen, 5 mg/kg, subcutaneous, Carporal; AST Farma BV). Mice were given eye cream (CAF; CEVA Sante Animale BW) and placed in a stereotactic frame (Kopf Instruments). Local anesthesia was applied by spraying lidocaine (100 mg/mL; Xylocaine, AstraZeneca BV), and two holes were drilled for entrance of the injection needles. AAV vectors, 500 nl, with a titer of $6.2 \times 10^{11}$ gc/ml were injected bilaterally (−2.46 mm posterior to bregma, +/− 2.2 mm lateral from bregma, and −1.3 mm ventral from the skull, under a 10˚ angle) at 100 nl per minute with a syringe pump (Harvard Apparatus) connected to stainless steel needles (31G, Coopers Needleworks) targeted to the CA1 region of the hippocampus. Needles were left in place for 10 minutes following the injection. After surgery, mice were given atipamezole (2.5 mg/kg, intraperitoneal, SedaStop; AST Farma BV) and saline for rehydration. During the following 7 days, mice continuously received carprofen through their drinking water (0.027 mg/ml).

After 4 weeks, acute brain slices were obtained. Mice were first anaesthetized with isoflurane and decapitated. Brains were rapidly isolated, and 250-μm-thick coronal slices were made on a vibratome (Leica VT1200 S) in ice-cold artificial cerebrospinal fluid (ACSF) containing (in mM) 124 NaCl, 26 NaHCO$_3$, 11 D-glucose, 2.5 KCl, 1 NaH$_2$PO$_4$, HEPES 5, 7 MgSO$_4$, and 0.5 CaCl$_2$. Subsequently, slices were transferred to an immersion-style holding chamber containing 124 NaCl, 26 NaHCO$_3$, 11 D-glucose, 2.5 KCl, 1 NaH$_2$PO$_4$, HEPES 5, 1 MgSO$_4$, and 2 CaCl$_2$, in which they recovered for at least 1 hour at RT. ACSF solutions were continuously bubbled with carbogen gas (95% O$_2$, 5% CO$_2$) and had an osmolarity of approximately 300 mOsm. After recovery, slices were stained for 1 hour with 250 nM Halo-JF646 ligand diluted in ACSF. Following rinsing with ACSF, slices were fixed overnight with 4% PFA, washed in PBS, and mounted with VectaShield (VectorLabs).

## Lentivirus production and infection

For lentivirus production, HEK293T cells were maintained at a high growth rate in DMEM supplemented with 10% FCS and 1% pen/strep. At 1 day after plating, cells were transfected using PEI (Polysciences) with second-generation LV packaging plasmids (psPAX2 and 2MD2. G) and a pFUGW construct containing the desired insert at a 1:1:1 molar ratio. At 6 hours after transfection, cells were washed once with PBS, and medium was replaced with DMEM containing 1% pen/strep. At 48 hours after transfection, the supernatant was harvested and briefly centrifuged at 700*g* to remove cell debris. The supernatant was concentrated using Amicon Ultra 15 100K MWCO columns (Milipore), and Cas9 and knock-in viruses were mixed at 1:1 and used immediately for infection. For cultured hippocampal neurons at DIV 2–4, 2–4 μl virus was added per well, and neurons were fixed at DIV 21–23 with 4% PFA/Suc for 10 minutes. For organotypic hippocampal slice cultures, virus was injected into the CA1 region at DIV 1 using an Eppendorf Femtojet injector. Slices were fixed at DIV 10 with 4% PFA in PBS for 30 minutes, washed 3 times for 10 minutes with PBS, and mounted with VectaShield (Vector Laboratories).

## Next-generation sequencing of genomic sites of integration

Genomic DNA was isolated from electroporated neurons at DIV 4. Neurons were lysed in lysis buffer (100 mM Tris, 50 mM EDTA, 40 mM NaCl, 0.2% SDS [pH 8.5]) and incubated with 100 μg/ml Proteinase K (Roche) at 55˚C for 2 hours, followed by 1 hour at 85˚C to inactivate Proteinase K. Genomic DNA was isolated by ethanol precipitation and dissolved in elution buffer (10 mM Tris [pH 8.0]) (Qiagen). Genomic PCR was performed to amplify the 5′ and 3′ junctions of the integrated donor (for PCR primers used, see S4 Table) using a touchdown PCR and Phusion HF polymerase (Thermo Fisher Scientific). Genomic primers were

designed using NCBI Primer-Blast. Knock-ins analyzed were primarily selected based on flanking genomic sequence, and we failed to amplify multiple alleles because of sequence complexity (e.g., sequence repeats, high GC content, or potential secondary structure). Amplicons were only included if they resulted in a well-resolved band on agarose gel. PCR products were separated using agarose gel electrophoresis and subsequently purified using a gel extraction kit (Qiagen). Purified PCR products were pooled with, on average, 10 ng per amplicon and sent for Illumina Miseq 2 × 300 bp next-generation sequencing (Utrecht Sequencing Facility [USEQ], Utrecht, the Netherlands).

Sequencing results were analyzed using CRIS.py [79]. Indel frequencies were plotted in a heatmap as the average percentage from the forward and reverse reads. The number of forward and reverse reads was averaged per junction for each knock-in and plotted. Indel and in-frame frequencies were also plotted compared with the Doench on-target score [30] and Bae out-of-frame score [31], respectively, obtained for each guideRNA sequence from UCSC genome browser gateway.

## Confocal imaging

Confocal images were acquired with a Zeiss LSM 700. For dissociated hippocampal cultures, neurons were imaged with a 63× NA 1.40 oil objective. A Z-stack containing 7–12 planes at a 0.56-μm interval was acquired with 0.1-μm pixel size, and maximum intensity projections were made for analysis and display. Organotypic and acute slices were imaged with a 20× NA 0.8 objective. Z-stacks were acquired with varying intervals. Image analysis was primarily performed using FIJI software [80]. Quantifications were performed in Excel 2016.

## gSTED superresolution imaging

Imaging was performed with a Leica TCS SP8 STED 3× microscope using an HC PL APO 100×/NA 1.4 oil immersion STED WHITE objective. The 488-nm wavelength of pulsed white laser (80 MHz) was used to excite Alexa488, the 561-nm to excite Alexa568, the 590-nm to excite Alexa594, and the 647-nm to excite Alexa647-, JF646-, and ATTO647N-labeled proteins. Alexa594, Alexa647, JF646, and ATTO647N were depleted with the 775-nm pulsed depletion laser, and we used an internal Leica HyD hybrid detector (set at 100% gain) with a time gate of $0.3 \leq tg \leq 6$ ns. Images were acquired as Z-stack using the 100× objective. Maximum intensity projections were obtained for image display and analysis.

In vivo STED images were additionally subjected to deconvolution using Huygens deconvolution software. Deconvolution was performed using the CMLE deconvolution algorithm, with a maximum of 40 iterations and the signal-to-noise ratio (SNR) set at 7.

## Quantification of knock-in efficiency

For quantification of knock-in efficiency over time, hippocampal neurons were transfected at DIV 3 with a 1:1 ratio mixture of pORANGE-β3-tubulin-GFP knock-in and pSM155-mCherry. Coverslips were fixed 24, 48, 72, 96, 120, and 144 hours after transfection using 4% PFA/Suc for 10 minutes at RT, washed three times with PBS/Gly, and mounted in Mowiol mounting medium.

For testing GFP-β-actin and GFP-GluN1 knock-in efficiencies, hippocampal neurons were transfected at DIV 3 with a 1:1 ratio mixture of pCaMK-Homer1c-mCherry overexpression construct together with pORANGE-GFP-β-actin #1 or #2 or pORANGE-GFP-GluN1 #1, #2, or #3 knock-in constructs. Neurons were fixed at DIV 21 using 4% PFA/Suc for 10 minutes at RT, washed three times with PBS/Gly, and mounted in Mowiol mounting medium.

Neurons were imaged with confocal microscopy as described above. For both experiments, mCherry- or Homer1c-mCherry-positive (i.e., transfected) neurons were manually counted and scored as being knock-in positive or negative. At least 1,000 transfected neurons from two independent neuronal cultures were scored for each time point or experimental condition.

### Quantification of synaptic PSD95 levels and enrichment and synapse size

Hippocampal neurons were transfected at DIV 3 with the pORANGE-PSD95-GFP knock-in construct or at DIV 15 with pSM155-PSD95-GFP overexpression plasmid [55] or pSM155-GFP [55]. At DIV 21, neurons were fixed and stained with mouse anti-PSD95 antibody 1:200 and Alexa594-conjugated secondary antibodies as described above. Neurons were imaged with confocal microscopy as described above. For each neuron, 50 circular regions of interest (ROI) of 1 μm in diameter were drawn around PSD95-GFP-positive synapses. For each ROI, the mean intensity of the GFP signal and anti-PSD95 staining was measured, background was subtracted, and values were normalized to the mean intensity value of all ROIs for both individual channels. Normalized intensity values for the PSD95-GFP knock-in signal and anti-PSD95 signal of individual synapses were plotted. In total, 550 synapses from 11 neurons divided over two independent neuronal cultures were used in the quantification.

To determine relative synaptic PSD95 content, PSD95 staining intensity in 22 circular ROIs of 1 μm in diameter around synapses per transfected (knock-in, overexpression, or GFP control) neuron was measured. Similarly, an equal number of ROIs were drawn around PSD95 puncta of nearby nontransfected neurons within the same image. Intensities of the anti-PSD95 channel were measured, and background was subtracted. Relative PSD95 content was quantified as the average anti-PSD95 intensity in synapses of a transfected neuron divided by those of the nontransfected neurons. To measure synapse size, a threshold was applied to the GFP signal (for PSD95-GFP knock-in and overexpression neurons) or anti-PSD95 signal (for GFP control), and individual synapses were detected using FIJI "Analyze Particles" with a detection size of 0.04-Infinity ($\mu m^2$) with a detection circularity of 0–1. Measured values were plotted as averages per analyzed neuron. To analyze synaptic enrichment of PSD95, circular ROIs were drawn within synapses and on the dendritic shaft. Mean GFP intensity was measured, background was subtracted, and values were averaged per neuron. Plotted ratio is the average intensity of synaptic GFP signal divided by that of the dendritic shaft. For each condition, at least 15 neurons from two independent neuronal cultures were analyzed.

To compare PSD95 levels in transfected but knock-in-negative neurons, neurons were transfected with a 1:1 ratio of pHomer1c-mCherry and the pORANGE empty vector or pHomer1c-mCherry and pPSD95-GFP knock-in construct at DIV 3. At DIV 21, neurons were fixed and stained for endogenous PSD95 as described above. Homer1c-mCherry-positive neurons were used to locate transfected neurons and to draw ROIs around synapses. For both conditions, 20 neurons from two independent neuronal cultures were analyzed.

### Quantification of Bassoon, Shank2, CaMKIIα, and F-actin levels

For Bassoon, neurons were transfected at DIV 3 with a 1:1 ratio of RIM1-HA under a synapsin promoter (overexpression construct) [81] and pORANGE template vector (control), pORANGE-GFP-Bassoon knock-in, or pORANGE-Bassoon-GFP knock-in. For Shank2, CaMKIIα, and β-actin, neurons were transfected with a 1:1 ratio of pHomer1c-mCherry (overexpression) and pORANGE template vector (control) or pORANGE-Shank2-GFP knock-in, pORANGE-GFP-CaMKIIα knock-in, pORANGE-GFP-β-actin knock-in #1, or pORANGE-GFP-β-actin knock-in #2. Neurons were stained as described above. For β-actin, the neurons were stained with Phalloidin-Alexa594 (Invitrogen) diluted 1:200 in blocking buffer for 1 hour at

RT. Coverslips were washed three times for 5 minutes in PBS/Gly and mounted in Mowiol mounting medium. For Bassoon, neurons were stained with anti-GFP (1:2,000) and anti-Bassoon (1:1,000) and anti-HA (1:200) antibodies as described above. For Shank2 and CaMKIIα, neurons were stained with anti-Shank2 (1:200) or anti-CaMKIIα (1:200) antibodies, respectively. Neurons were imaged with confocal microscopy as described above. Per transfected neuron, both knock-in positive and negative, 20 circular ROIs of 1 μm in diameter were manually drawn around synapses based on Homer1c or RIM signal. Similarly, an equal number of ROIs were drawn around puncta of nearby nontransfected neurons within the same image based on the antibody staining. To measure relative protein levels, antibody or phalloidin labeling intensities of individual ROI measurements were background subtracted and averaged for each neuron. The average intensity in the transfected neuron relative to the nontransfected neuron from the same image is plotted. For each condition, between 10 and 18 neurons from at least two independent neuronal cultures were analyzed.

## Live-cell imaging of β-actin dynamics

Imaging was performed on a spinning disk confocal system (CSU-X1-A1; Yokogawa) mounted on a Nikon Eclipse Ti microscope (Nikon) with Plan Apo VC 100× 1.40 NA oil objective (Nikon) with excitation from Cobolt Calyspso (491 nm) and emission filters (Chroma). The microscope was equipped with a motorized XYZ stage (ASI; MS-2000), Perfect Focus System (Nikon), and Prime BSI sCMOS camera (Photometrics) and was controlled by MetaMorph software (Molecular Devices). Neurons were maintained in a closed incubation chamber (Tokai hit: INUBG2E-ZILCS) at 37°C in 5% $CO_2$ in 200 μl of conditioned medium.

For studying actin dynamics upon Jasp treatment, neurons were transfected with pORANGE-GFP-β-actin knock-in #2 construct at DIV 3 and imaged at DIV 21–23 on a spinning disk confocal system (described above). Every 1 minute, a Z-stack was obtained in a range of 5.5 μm (12 planes with 0.5-μm intervals). After 5 minutes baseline imaging (6 frames), 100 μl/30 μM of Jasp (10 μM final concentration) or DMSO diluted in conditioned medium was added to the incubation chamber. Imaging was continued for another 20 minutes (21 frames) after addition. For analysis, maximum intensity projections were obtained, and drift was corrected. Background was subtracted in FIJI software using a rolling ball radius of 3.15 μm. For each neuron, four ROIs of variable sizes containing at least one spine each were drawn. Integrated densities (InDen) were measured for each frame. Frame-to-frame differences were obtained by subtracting each frame ($t_x$) from the previous ($t_{x-1}$) using a macro developed by Jacob Pruess. Frame-to-frame differences of the selected ROIs were measured and subtracted from the InDen at $t_x$ and normalized to the InDen $t_x$ to obtain the frame-to-frame correlation for each ROI at each time point, such that correlation = (InDen $t_x$−[InDen $t_x$−InDen $t_{x-1}$])/InDen $t_x$. Frame-to-frame correlation was plotted over time. For statistical analysis, the frame-to-frame correlation of the last five time points for each ROI was averaged per cell. For each condition, measurements from 28 ROIs from seven neurons divided over two independent neuronal cultures were used in the analysis.

For FRAP experiments, neurons were transfected with the GFP-β-actin knock-in #2 construct at DIV 3 and imaged at DIV 21–23 on a spinning disk confocal system (described above). FRAP experiments were performed using the ILas2 system (Roper Scientific). Experiments were performed in the presence of 10 μM Jasp or DMSO added to the imaging chamber 5 minutes before the start of the acquisition. After 2 minutes baseline imaging (single Z-plane, five frames with 30-second intervals), six ROIs with a fixed diameter of 1.26 μm containing dendritic spines were bleached using a targeted laser. Imaging during fluorescence recovery was continued for 5 minutes (13 frames with 10-second intervals followed by six frames with

30-second intervals). For analysis, acquisitions were corrected for drift. For each ROI, mean intensities were measured for every time point and corrected for background using the averaged intensity of two background ROIs. For each ROI, intensities were normalized to the averaged intensities of the frames before bleaching and normalized to zero based on the intensity from the first frame after bleaching. Normalized intensities were plotted over time. The mobile fraction of protein was calculated by averaging the normalized intensity of the last five frames for each neuron. For each condition, five neurons divided over two independent neuronal cultures were used in the analysis.

## Preparation of dissociated hippocampal cultures for gSTED

Hippocampal neurons were transfected with indicated knock-in constructs at DIV 3 and fixed at DIV 21. Dual-color gSTED imaging (as described above) was performed on PSD95-GFP, GFP-β-actin #1, GFP-GluN1 #1, GFP-CaMKIIα, GSG1L-GFP, and FRRS1L-GFP knock-in neurons stained with anti-GFP and anti-PSD95. pORANGE FRRS1L-GFP was cotransfected with pSyn tagRFP-ER [82]. (Dual-color) gSTED imaging was additionally performed on extracted cytoskeleton of the GFP-β-actin and β3-tubulin-GFP knock-in neurons. At DIV 7 (β3-tubulin-GFP knock-in) and DIV 21 (GFP-β-actin knock-in), the neuronal cytoskeleton was extracted using extraction buffer (PEM80-buffer [80 mM PIPES, 1 mM EGTA, 2 mM MgCl$_2$ (pH 6.9)], 0.3% Triton-X, 0.1% glutaraldehyde) for 1 minute at RT. Next, neurons were fixed with PFA/Suc for 10 minutes at RT, washed three times for 5 minutes with PBS/Gly, and subsequently incubated with 1 mg/ml sodium borohydride in PBS for 7 minutes at RT. Coverslips were washed 3 times for 5 minutes with PBS/Gly. The GFP signal was enhanced with anti-GFP staining. The β3-tubulin-GFP knock-in was additionally stained for α-tubulin diluted 1:1,000. Anti-GFP primary antibodies were stained with Alexa488- or ATTO647N-conjugated secondary antibodies, and anti-PSD95 and anti-α-tubulin were stained with the Alexa594- or ATTO647N-conjugated secondary antibody (all as described above). To label surface receptors, GFP-GluN1 knock-in neurons were stained with anti-GFP prior to permeabilization and subsequent anti-PSD95 staining.

## Quantification of colocalization gSTED

Using ImageJ software, a line scan of interest was drawn to obtain pixel intensity data to assess the degree of colocalization between two structures along that line. To quantify the degree of colocalization between two structures, entire images showing parts of the dendritic tree of a knock-in neuron were used for analysis. First, all dendritic spines (positive for both proteins: PSD95-GFP knock-in and anti-PSD95 staining or GFP-β-actin knock-in and anti-PSD95 staining) were selected by drawing ROIs in ImageJ. Next, the ROIs were combined to clear the outside of the ROIs to remove all background from surrounding neurons or dendritic shafts. Then, the ImageJ plug-in "JaCoP" (Just Another Colocalization Plug-in) was used to calculate the PCC and MOC. For the MOC, the thresholding was done manually. These analyses were performed on both the confocal and STED maximum projections of the exact same regions (of a neuron). In total, 10 PSD95-GFP knock-in and seven GFP-β-actin knock-in neurons were analyzed from two independent experiments.

## Confocal and STED quantifications of NMDA receptors

Neurons were transfected at DIV 3 with the pORANGE-GFP-GluN1 knock-in #1 construct. Neurons were fixed at DIV 21 and stained with anti-PSD95 as described above. Neurons were imaged with confocal microscopy as described above. For each neuron, 50 circular ROIs of 1 μm in diameter were drawn around GFP-GluN1-positive synapses. For each ROI, the mean

intensity of the GFP signal and anti-PSD95 staining was measured, background was subtracted, and values were normalized to the mean intensity value of all ROIs for both individual channels. Normalized intensity values for the GFP-GluN1 knock-in signal and anti-PSD95 signal of individual synapses were plotted. In total, 450 synapses from nine neurons divided over two independent neuronal cultures were used in the analysis.

The FIJI plug-in Full Width at Half Maximum (FWHM) macro developed by John Lim was used to measure the FWHM from intensity profiles using Gaussian fitting. Line scans were drawn along the width and length of identified GluN1 substructures (by setting an appropriate brightness/contrast) to obtain the FWHM of the length and width of these substructures. Subsequently, these substructures were categorized as synaptic or extrasynaptic based on the colocalization with PSD95. For image display, the length was plotted against the width for each cluster. In all, 479 GFP-GluN1 clusters (387 synaptic, 92 extrasynaptic) from three neurons were analyzed.

For the quantification of total GluN1 cluster area per synapse, and correlation with synapse area, the same images were used as for the quantification of the FWHM of the GluN1 substructures. Specifically, the STED resolved images were used for the quantification of GluN1 cluster area, whereas the confocal images were used to quantify the area of the PSD, using PSD95 as a marker. First, an ROI was drawn around the knock-in neuron of interest to clear the outside of the ROI, removing all background. Subsequently, the image was subjected to thresholding to isolate the objects of interest from the background and watershedding to separate overlapping objects. Then, all objects (GluN1 clusters and PSDs) were detected using "Analyze Particles" with a detection size of 0.02-Infinity ($\mu m^2$) for GluN1 substructures and 0.04-Infinity ($\mu m^2$) for PSDs, and all with a detection circularity of 0–1.

## SMLM and detection

dSTORM imaging was performed on a Nikon Ti microscope equipped with a Nikon 100× NA 1.49 Apo total internal reflection fluorescence (TIRF) oil objective, a Perfect Focus System. Effective pixel size is 65 nm. Oblique laser illumination was achieved using a custom illumination pathway with a 60-mW, 405-nm-diode laser (Omicron); a 200-mW, 491-nm-diode laser (Omicron); and a 140-mW, 641-nm-diode laser (Omicron). Emission light was separated from excitation light with a quad-band polychroic mirror (ZT405/488/561/640rpc, Chroma) and additional band-pass emission filters (ET 525/595/700, Chroma). Fluorescence emission was acquired using an ORCA-Flash 4.0v2 CMOS camera (Hamamatsu). Lasers were controlled using Omicron software, whereas all other components were controlled by µManager software [83].

Live-cell SMLM imaging experiments were performed on a Nikon Ti microscope equipped with a 100× NA 1.49 Apo TIRF oil objective, a Perfect Focus System, and an additional 2.5× Optovar to achieve an effective pixel size of 64 nm. Oblique laser illumination was achieved using a custom illumination pathway with an AA acousto-optic tunable filter (AA opto-electronics); a 15-mW, 405-nm-diode laser (Power Technology); a 100-mW, 561-nm-DPSS laser (Cobolt Jive); and a 40-mW, 640-nm-diode laser (Power Technology). Emission light was separated from excitation light with a quad-band polychroic mirror (ZT405/488/561/640rpc, Chroma) and additional band-pass emission filters (ET 525/595/700, Chroma). Fluorescence emission was acquired using a DU-897D EMCCD camera (Andor). All components were controlled by µManager software [83].

Acquired image stacks were analyzed using the ImageJ plug-in Detection of Molecules (DoM) v1.1.5 [84]. Briefly, each image was convoluted with a 2D Mexican hat–type kernel that matches the microscope's point spread function. Spots were detected by thresholding the

images and localized by fitting a 2D Gaussian function using unweighted nonlinear least-squares fitting with the Levenberg–Marquardt algorithm. Drift correction was applied by calculating the spatial cross-correlation function between intermediate superresolved reconstructions.

## Single-molecule tracking PALM and analysis

Neurons were transfected with the mEos3.2-CaMKIIα knock-in construct at DIV 3 and imaged at DIV 21–23. Neurons were imaged in extracellular imaging buffer (10 mM HEPES, 120 mM NaCl, 3 mM KCl, 2 mM CaCl$_2$, 2 mM MgCl$_2$, 10 mM glucose [pH 7.35]) at RT. mEos3.2 molecules were photoconverted from green to red fluorescence using simultaneous 405-nm and 561-nm illumination using TIRF. Stacks of 5,000–7,000 frames were acquired at 50 Hz. PALM reconstruction was made in DoM, plotting localizations based on their localization precision, rendered with a pixel size of 10 × 10 nm. Molecules localized with precision <25 nm were used for further analysis. Tracking was accomplished using custom tracking algorithms in MATLAB (MathWorks) using a tracking radius of 512 nm. For tracks consisting of ≥4 frames, the instantaneous diffusion coefficient was estimated as described [40]. The first three points of the MSD versus elapsed time (*t*) plot were used to fit the slope using linear fitting adding a value of 0 at MSD(0). Tracks with a negative slope (<8%) were ignored. The diffusion coefficient $D_{eff}$ was then calculated using MSD = $4D_{eff}\,t$. Individual tracks were plotted using MATLAB, and each was given a random color. All single-molecule trajectories from all acquisitions were used to visualize a frequency distribution. On this, we fitted two Gaussian distributions to identify the two kinetic populations. Mean values for the two fits were calculated per analyzed neuron and plotted. In total, 11 neurons from two independent experiments were included in the analysis.

## dSTORM imaging and analysis

Hippocampal neurons were transfected at DIV 3 with the GFP-GluN1 knock-in construct #1 and fixed on DIV 21. Neurons were surface stained with anti-GFP 1:2,000 and Alexa647-conjugated secondaries as described above. Neurons were postfixed in 4% PFA/Suc for 5 minutes, additionally washed 3 times with PBS/Gly, and kept in PBS at 4°C until imaging. dSTORM imaging was performed in PBS containing 10–50 mM MEA, 5% w/v glucose, 700 μg/ml glucose oxidase, and 40 μg/ml catalase. GFP-GluN1 knock-in-positive neurons were located on GFP signal. For dSTORM, the sample was illuminated (in TIRF) with continuous 647-nm laser light and gradually increasing intensity of 405-nm laser light. Stacks of 10,000–15,000 frames were acquired at 50 Hz. dSTORM reconstruction was made in DoM, plotting localizations based on their localization precision, rendered with a pixel size of 10 × 10 nm. Molecules with a localization precision <15 nm were selected for further analysis. Next, blinking events longer than one frame were filtered out by tracking (tracking radius of 130 nm). GluN1 clusters were identified using the DBSCAN algorithm [62] implemented in MATLAB. Subsequently, the alpha shape was used as the cluster border. Clusters with a density of >5,000 molecules per micrometer were used for further analysis. For each individual cluster, molecules were plotted and color-coded according to the local density [55], defined as the number of molecules within a radius of 5 times the mean nearest neighbor distance of all molecules within the cluster. Molecules with a local density value >40 were considered to be enriched in a nanodomain. Nanodomains were isolated using MATLAB functions linkage() and cluster(). The polygon circumventing molecules belonging to individual nanodomains was used to calculate the diameter of the nanodomain. Nanodomains containing <5 localizations and

diameter <30 nm were rejected. In total, 859 clusters from three neurons from two independent experiments were analyzed.

## uPAINT and analysis

Neurons transfected with the GFP-GluN1 knock-in construct #1 and pCamk Homer1c-mCherry at DIV 3 were imaged at DIV 21–23 in extracellular imaging buffer supplemented with 0.8% BSA. GFP-GluN1-positive neurons were identified by GFP signal, and ATTO647N-conjugated anti-GFP nanobodies (GFPBooster-ATTO647N, Chromotek) were bath applied to a final dilution of 1:50,000. Imaging was conducted at a 50-Hz frame rate with 640-nm excitation laser illumination (in TIRF). Molecules fitted with a precision <50 were tracked with tracking radius of 512 nm and diffusion coefficient determined for tracks >30 frames. A cell mask was drawn manually to filter out localizations outside neurons due to nonspecifically bound nanobody. Tracking and estimation of the instantaneous diffusion was performed as described for the PALM imaging. Synapses were identified based on widefield Homer1c-mCherry signal as described [85]. Synaptic tracks were defined as tracks in which 80% of the localizations were located within the border of the synapse. All others were considered extrasynaptic. In total, 6 neurons from three independent experiments were analyzed.

## Statistics

Statistical significance was tested with a Student $t$ test when comparing two groups. A $P$ value below 0.05 was considered significant. If multiple groups were compared, statistical significance was tested with a one-way ANOVA followed by a Bonferroni's multiple comparison. In all figures, * was used to indicate a $P$ value < 0.05, ** for $P < 0.01$, and *** for $P < 0.001$. Reported $n$ is number of neurons, and each experiment was replicated in neuronal cultures from at least two independent preparations. Statistical analysis and graphs were prepared in GraphPad Prism, and figures were generated in Adobe Illustrator CC.

## Additional resources

Plasmids from this study will be made available through Addgene (see S5 Table).

## Supporting information

**S1 Fig. Schematic of knock-in construct design (related to Fig 1).** (A and B) Examples of knock-in construct design for *Gria1* (A) and *Dlg4* (B), which contain target sequences in opposite genomic strands. The target sequence is indicated in blue, the PAM sequence is in red, and the part of the primer used for PCR amplification of the donor DNA is shown in yellow. Amino acid sequence is shown under the sequences. Asterisk indicates stop codon. Red dotted lines indicate position of Cas9 cleavage and sites of integration. Purple line indicates restriction enzyme sites used for cloning into pORANGE. Dlg4, Discs Large MAGUK Scaffold Protein 4; Gria1, glutamate ionotropic receptor AMPA type subunit 1; ORANGE, Open Resource for the Application of Neuronal Genome Editing; PAM, protospacer adjacent motif.
(TIF)

**S2 Fig. ORANGE CRISPR/Cas9 knock-in library (related to Fig 2).** Representative images of cultured hippocampal knock-in neurons. Examples shown are used for zooms shown in Fig 2D. DIV 21. Asterisk indicates signal enhanced using anti-GFP antibodies (Alexa488 or Alexa647). Scale bar, 5 μm. GFP, green fluorescent protein; ORANGE, Open Resource for the Application of Neuronal Genome Editing.
(TIF)

**S3 Fig. Localization of ORANGE knock-ins relative to synaptic makers (related to Fig 2).**
(A) Examples of GFP knock-in (green) relative to anti-Bassoon staining (magenta, Alexa647)
as presynaptic marker or (B) anti-PSD95 staining (magenta, Alexa647) as postsynaptic marker
in cultured hippocampal neurons. Asterisk indicates signal enhancement using anti-GFP anti-
bodies (Alexa488). Scale bars, 5 μm. Arrows indicate examples of GFP-positive objects. GFP,
green fluorescent protein; ORANGE, Open Resource for the Application of Neuronal Genome
Editing.
(TIF)

**S4 Fig. ORANGE knock-ins in dissociated neuronal culture and organotypic slices using a
dual-lentiviral approach (related to Fig 3).** (A) Overview of lentiviral constructs and timeline
showing age of infection and fixation. (B) Representative images of infected (magenta) pri-
mary rat hippocampal neurons positive for GluA1-GFP knock-in or β3-tubulin-GFP knock-in
(green). Scale bars, 20 μm and 5 μm for the overview and zooms, respectively. (C) Representa-
tive images of GluA1-GFP knock-in in organotypic hippocampal slices from mice. Shown are
a series of individual 1-μm planes from a Z-stack. Arrows indicate GFP-positive cells. Scale
bar, 20 μm. (D) Representative zooms of GluA1-GFP knock-in dendrites from a CA1 pyrami-
dal cell and an aspiny interneuron. Shown are individual 0.5-μm planes from a Z-stack and the
maximum projection (max). Scale bar, 2 μm. CA1, cornu ammonis region 1; GFP, green fluo-
rescent protein; GluA1, Glutamate receptor AMPA 1; ORANGE, Open Resource for the
Application of Neuronal Genome Editing.
(TIF)

**S5 Fig. Efficiency of ORANGE knock-in over time in cultured neurons (related to Fig 4).**
(A) Schematic overview of knock-in and mCherry reporter plasmids and (B) experimental
setup. (C) Representative images of β3-tubulin-GFP knock-in (green) cotransfected with an
mCherry fill (magenta) fixed 24 hours (DIV 4) and 144 hours (DIV 9) after transfection. Scale
bar, 20 μm. (D) Quantification of β3-tubulin-GFP knock-in efficiency over time as percentage
of transfected (mCherry-positive) neurons. Data are represented as means ± SEM. Underlying
data can be found in S1 Data. DIV, day in vitro; GFP, green fluorescent protein; ORANGE,
Open Resource for the Application of Neuronal Genome Editing.
(TIF)

**S6 Fig. Next-generation sequencing of donor integration at targeted locus (related to Fig
4).** (A) Schematic overview of experimental setup. Neurons were electroporated immediately
after dissociation and cultured until DIV 4. Genomic DNA was isolated, and the 5′ and 3′ junc-
tions of integration were amplified with PCR, pooled, and subjected to next-generation
sequencing. (B) Heatmap summarizing the sequencing results for 5′ and 3′ junction amplicons
of the indicated knock-ins. Heatmap is color-coded for the frequency of indel size, as analyzed
using CRIS.py. For a few genes, we were only able to amplify one of the two junctions with
PCR. (C) Average number of reads obtained with deep sequencing for all successfully analyzed
knock-ins (mean 5′: $1.69 \times 10^5$ reads $\pm 0.18 \times 10^5$, 3′: $1.57 \times 10^5 \pm 0.16 \times 10^5$). (D) Accuracy of
knock-in plotted for each junction. Plotted points indicate percentage of zero indels from all
knock-ins in (B) (mean 5′: 54.2% ± 7.0%, 3′: 60.7% ± 5.4%). Green points indicate minor
mutations that do not influence the reading frame for this particular integration (e.g., frame
shift after stop codon). (E) Correlation graph between zero indel frequency per amplicon and
Doench on-target score of the gRNA target sequence. (F) Correlation graph between correct
reading frame integration frequency and Bae out-of-frame score of the gRNA target sequence.
Data are presented as means ± SEM. Underlying data can be found in S1 Data. DIV, day in

vitro; gRNA, guide RNA.
(TIF)

**S7 Fig. Comparison of integration efficiency at different PAM sites in the same gene
(related to Fig 4).** (A and E) Genomic regions of the *Actb* and *Grin1* genes around the targeted integration site are shown. The PAM (red line and red shaded boxes) and target sequences (blue) are shown below for each of the tested knock-ins. Intron sequences are in lowercase, and exon sequences are in uppercase. Additional protein information is shown above the sequence. (B and F) Tables containing information about the site of integration at the protein level and MIT, Doench, and Bae scores of the individual guide RNA sequences (determined based on rat genomic sequence from the UCSC RGSC5.0/rn5 genome assembly). (C and G) Representative images of neurons transfected with the various β-actin (C) or GluN1 (G) knock-in constructs targeting the PAM sites shown in (A) and (E), respectively. Scale bars, 10 μm and 2 μm for the overviews and zooms, respectively. (D and H) Knock-in efficiency determined as the percentage of GFP-β-actin (D) or GFP-GluN1 (H)-positive neurons coexpressing Homer1c-mCherry. Data are presented as means ± SEM. * $P < 0.05$, *** $P < 0.001$, ANOVA or Student *t* test. Underlying data can be found in S1 Data. Actb, Actin Beta; GFP, green fluorescent protein; GluN1, Glutamate receptor NMDA 1; Grin1, glutamate ionotropic receptor NMDA type subunit 1; PAM, protospacer adjacent motif; UCSC, University of California Santa Cruz.
(TIF)

**S8 Fig. Quantification of Bassoon, Shank2, CaMKIIα, and F-actin levels in knock-in neurons (related to Fig 4).** (A) Representative images of neurons transfected with Homer1c-mCherry (cyan) together with pORANGE empty vector as control or Shank2-GFP knock-in (green). Neurons were stained with anti-Shank2 (magenta, Alexa647). (C) Images of neurons transfected with Homer1c-mCherry (cyan) together with pORANGE empty vector as control GFP-CaMKIIα knock-in (green). Neurons were stained with anti-CaMKIIα (magenta, Alexa647). (E) Neurons transfected with Homer1c-mCherry together with pORANGE empty vector as control, GFP-β-actin knock-in #1 or GFP-β-actin knock-in #2 (green). Neurons were stained for F-actin using phalloidin (magenta, Alexa647). Scale bar, 5 μm. (B, D, and F) Quantification of protein levels relative to transfected neurons. (B) Relative fluorescence intensity: control: 0.84 ± 0.04, *n* = 16 neurons, Shank2-GFP knock-in: 0.67 ± 0.07, *n* = 17 neurons, $P > 0.05$, knock-in negative: 0.20 ± 0.04, *n* = 17 neurons, $P < 0.01$, ANOVA. (D) Control: 1.04 ± 0.03, *n* = 6 neurons, GFP-CaMKIIα knock-in: 0.88 ± 0.05, *n* = 11 neurons, $P > 0.05$, knock-in negative: 0.82 ± 0.09, *n* = 13 neurons, $P > 0.05$, ANOVA. (F) Control: 1.00 ± 0.03, *n* = 10 neurons, GFP-β-actin knock-in #1: 1.01 ± 0.03, *n* = 12 neurons, $P > 0.05$, knock-in #1 negative: 1.01 ± 0.04, *n* = 8 neurons, $P > 0.05$, GFP-β-actin knock-in #2: 1.01 ± 0.03, *n* = 12 neurons, $P > 0.05$, knock-in #2 negative: 0.92 ± 0.04, *n* = 7 neurons, $P > 0.05$, ANOVA. Data are presented as means ± SEM. *** $P < 0.001$, ANOVA. Underlying data can be found in S1 Data. CaMKIIα, Calcium/calmodulin-dependent protein kinase type II subunit alpha; GFP, green fluorescent protein; ns, not significant; ORANGE, Open Resource for the Application of Neuronal Genome Editing; SHANK2, SH3 and multiple ankyrin repeat domains protein 2.
(TIF)

**S9 Fig. Live-cell superresolution PALM imaging of endogenous CaMKIIα dynamics (related to Fig 5).** (A) Example of a dendrite expressing mEos3.2-CaMKIIα knock-in. Scale bar, 2 μm. (B) Single-molecule PALM reconstruction of dendrite shown in (A). Scale bar, 2 μm. (C) Individual single-molecule trajectories. Scale bar, 2 μm. Dotted line indicates cell outline. (D) Representative zooms of single-molecule trajectories in individual spines. Scale

bar, 200 nm. (E) Frequency distribution of diffusion coefficients derived from single-molecule trajectories (black line). Mixed Gaussian fits (red and blue) indicate two kinetic populations. (F) Quantification of mean diffusion coefficient for each of the two kinetic populations. Data are presented as means ± SEM. Underlying data can be found in S1 Data. CaMKIIα, Calcium/calmodulin-dependent protein kinase type II subunit alpha; mEos3.2, monomeric Eos 3.2; PALM, photoactivated localization microscopy.
(TIF)

**S10 Fig. gSTED imaging and colocalization analysis (related to Fig 6).** (A) Representative gSTED overview image and zooms (B) of β3-tubulin-GFP knock-in neurons (DIV 7) extracted and stained with anti-GFP (green, ATTO647N) and anti-α-tubulin (magenta, Alexa594). Scale bars, 20 μm, 4 μm, and 2 μm for the overview and zooms, respectively. (C) Intensity profile along the line indicated in (B). (D) PCC quantifying colocalization between PSD95-GFP or GFP-β-actin knock-in signal with anti-PSD95 staining intensity. Related to Fig 6F–6K. (E and F) Manders' correlation of PSD95-GFP or GFP-β-actin knock-in overlapping with PSD95 staining (*M1*) (E) or anti-PSD95 staining overlapping with PSD95-GFP or GFP-β-actin knock-in (*M2*) (F) related to Fig 6F–6K. Average values: (PSD95, confocal: median PCC = 0.95, *M1* = 0.78, *M2* = 0.80, STED: PCC = 0.88, ANOVA, $P < 0.001$, *M1* = 0.62, $P < 0.001$, *M2* = 0.71, $P > 0.05$, n = 10 neurons) (β-actin, confocal: median PCC = 0.88, $P < 0.001$, *M1* = 0.50, $P < 0.001$, *M2* = 0.74, $P > 0.05$, STED: median PCC = 0.78, ANOVA, $P < 0.001$, *M1* = 0.18, $P < 0.001$, *M2* = 0.48, $P < 0.001$, $n = 7$ neurons). (G and H) gSTED of dendrites and zooms positive for GFP-CaMKIIα knock-in stained with anti-GFP (green, ATTO647N) and anti-PSD95 (magenta, Alexa594). Scale bars, 2 μm and 500 nm for (G) and (H), respectively. (I and J) Line scans of individual spines indicated in (H). Data are represented as means ± SEM. ***$P < 0.001$. ANOVA. Underlying data can be found in S1 Data. CaMKIIα, Calcium/calmodulin-dependent protein kinase type II subunit alpha; DIV, day in vitro; GFP, green fluorescent protein; gSTED, gated stimulated-emission depletion; ns, not significant; PCC, Pearson's correlation coefficient; PSD95, postsynaptic density protein 95.
(TIF)

**S11 Fig. Multiplex labeling using CAKE (related to Fig 8).** (A) Mechanism of sequential knock-in activation using CAKE. (B) Example of correct dual-color labeling (top) and incorrect dual-color labeling (bottom) with GluA1-GFP-P2A-Cre and Lox Halo β-actin. Arrowheads indicate dendritic spines, and arrow indicates the axon. (C) Example of correct dual-color labeling (top) and incorrect dual-color labeling (bottom) with β3-tubulin-GFP-P2A-Cre and Lox GluA1-HA. Scale bar is 10 μm for overview and 5 μm for zooms. CAKE, conditional activation of knock-in expression; GFP, green fluorescent protein; GluA1, Glutamate receptor AMPA 1; HA, hemagglutinin.
(TIF)

**S1 Table. Design rationale for each ORANGE knock-in.** ORANGE, Open Resource for the Application of Neuronal Genome Editing.
(DOCX)

**S2 Table. Target sequences for ORANGE knock-ins.** ORANGE, Open Resource for the Application of Neuronal Genome Editing.
(DOCX)

**S3 Table. Donor PCR primers.**
(DOCX)

**S4 Table. Genomic PCR primers.**
(DOCX)

**S5 Table. Overview ORANGE constructs with Addgene IDs.** ORANGE, Open Resource for the Application of Neuronal Genome Editing.
(DOCX)

**S1 Data. Raw data.**
(XLSX)

## Acknowledgments

We want to thank all members of the MacGillavry lab for discussions. We want to thank Corette J. Wierenga for providing organotypic hippocampal slice cultures and Onur Basak for helpful discussions. We want to thank Anna Akhmanova, Lukas C. Kapitein, and Corette J. Wierenga for critical reading of the manuscript.

## Author Contributions

**Conceptualization:** Jelmer Willems, Arthur P. H. de Jong, Nicky Scheefhals, Harold D. MacGillavry.

**Formal analysis:** Jelmer Willems, Arthur P. H. de Jong, Nicky Scheefhals, Harold D. MacGillavry.

**Funding acquisition:** Harold D. MacGillavry.

**Investigation:** Jelmer Willems, Arthur P. H. de Jong, Nicky Scheefhals, Eline Mertens, Lisa A. E. Catsburg, Rogier B. Poorthuis, Harold D. MacGillavry.

**Methodology:** Jelmer Willems, Arthur P. H. de Jong, Nicky Scheefhals, Harold D. MacGillavry.

**Resources:** Fred de Winter, Joost Verhaagen, Frank J. Meye, Harold D. MacGillavry.

**Supervision:** Harold D. MacGillavry.

**Validation:** Jelmer Willems, Arthur P. H. de Jong, Nicky Scheefhals, Harold D. MacGillavry.

**Visualization:** Jelmer Willems, Arthur P. H. de Jong, Nicky Scheefhals, Harold D. MacGillavry.

**Writing – original draft:** Jelmer Willems, Arthur P. H. de Jong, Nicky Scheefhals, Harold D. MacGillavry.

**Writing – review & editing:** Jelmer Willems, Arthur P. H. de Jong, Nicky Scheefhals, Harold D. MacGillavry.

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
