## [Editor Report · Decision Letter 0]

30 Jul 2019

Dear Harold, 

Thank you for submitting your manuscript entitled "ORANGE: A CRISPR/Cas9-based genome editing toolbox for epitope tagging of endogenous proteins in neurons" for consideration as a Methods and Resources article by PLOS Biology.

Your manuscript has now been evaluated by the PLOS Biology editorial staff, as well as by an Academic Editor with relevant expertise, and I am writing to let you know that we would like to send your submission out for external peer review. Please accept my apologies for the delay in sending this initial decision to you.

*Please be aware that, due to the voluntary nature of our reviewers and academic editors, manuscripts may be subject to delays during the holiday season. Thank you for your patience.*

**Important**: Please also see below for further information regarding completing the MDAR reporting checklist. The checklist can be accessed here: https://plos.io/MDARChecklist

Please re-submit your manuscript and the checklist, within two working days, i.e. by Aug 01 2019 11:59PM.

Kind regards,

Gabriel Gasque, Ph.D.,

Senior Editor

PLOS Biology

INFORMATION REGARDING THE REPORTING CHECKLIST:

PLOS Biology is pleased to support the "minimum reporting standards in the life sciences" initiative (https://osf.io/preprints/metaarxiv/9sm4x/). This effort brings together a number of leading journals and reproducibility experts to develop minimum expectations for reporting information about Materials (including data and code), Design, Analysis and Reporting (MDAR) in published papers. We believe broad alignment on these standards will be to the benefit of authors, reviewers, journals and the wider research community and will help drive better practise in publishing reproducible research. 

We are therefore participating in a community pilot involving a small number of life science journals to test the MDAR checklist. The checklist is intended to help authors, reviewers and editors adopt and implement the minimum reporting framework. 

IMPORTANT: We have chosen your manuscript to participate in this trial. The relevant documents can be located here:

MDAR reporting checklist (to be filled in by you): https://plos.io/MDARChecklist

**We strongly encourage you to complete the MDAR reporting checklist and return it to us with your full submission, as described above. We would also be very grateful if you could complete this author survey:

https://forms.gle/seEgCrDtM6GLKFGQA

Additional background information:

Interpreting the MDAR Framework: https://plos.io/MDARFramework

Please note that your completed checklist and survey will be shared with the minimum reporting standards working group. However, the working group will not be provided with access to the manuscript or any other confidential information including author identities, manuscript titles or abstracts. Feedback from this process will be used to consider next steps, which might include revisions to the content of the checklist. Data and materials from this initial trial will be publicly shared in September 2019. Data will only be provided in aggregate form and will not be parsed by individual article or by journal, so as to respect the confidentiality of responses. 

Please treat the checklist and elaboration as confidential as public release is planned for September 2019.

We would be grateful for any feedback you may have.

---

## [Decision Letter · Decision Letter 1]

11 Sep 2019

Dear Harold,

Thank you very much for submitting your manuscript "ORANGE: A CRISPR/Cas9-based genome editing toolbox for epitope tagging of endogenous proteins in neurons" as a Methods and Resources article for review by PLOS Biology. Your paper has now been assessed and discussed by the PLOS Biology editors, by an Academic Editor with relevant expertise, and by four independent reviewers. Based on the reviews, I regret that we will not be pursuing this manuscript for publication in the journal. Please accept my sincere apologies for the delay in sending this decision to you.

As you will see, the reviewers agree that the resource, to be of significant use to the community, still requires additional validation and further implementation. Unfortunately, we feel that to meet our editorial standards, this additional work is needed, and we have deemed it to be too extensive to invite a revision.

The reviews are attached, and we hope they may help you should you decide to revise the manuscript for submission elsewhere. I am sorry that we cannot be more positive on this occasion, but I hope you appreciate the reasons for this decision.

While we cannot consider your manuscript for publication in PLOS Biology, I suggest you consider submitting it to PLOS ONE. If you are interested in this option, please let me know, and I might be able to transfer your reviews.

I hope you will consider PLOS Biology for other submissions in the future. Thank you for your support of PLOS and of Open Access publishing.

Sincerely,

Gabriel Gasque, Ph.D., 

Senior Editor

PLOS Biology

Reviewer remarks:

Reviewer #1: In general, the authors established a very good tagging toolkit for monitoring the endogenous protein by using HITI-CRISPR techniques for taking the advantages of HITI can work with higher efficiency in post-mitotic cells. With the combination of the high resolution of gSTED microscopy images, ORANGE Toolkit indeed provides a powerful approach to monitor the expression and dynamics in living cells, especially for those proteins without good antibodies.

There are some minor comments regarding this manuscript:

1. For HITI-CRISPR precise KI analysis, it is strongly suggested that only analyze the eGFP-positive cells instead of analyze a pool of cells. A N-tag of eGFP might cause out-of-frame for the coding sequences of target gene, when the other allele is also edited, a KO cell was generated. If the target region is not analyzed in sorted cells (i.e. eGFP positive cells), the aberrant expression pattern cannot be ruled out unless a specific antibody was used parallely to double check the pattern. 

2. In Figure S3-D, due to duplication of incomplete target sequences (17 bases, 5’tggcactgggcaggcaggg) in the 5’juntion in GFP-CamKIIa, a new ATG start codon was generated. This new ATG-frame will cause an out-of-frame reading to the subsequent eGFP coding sequence. To overcome this situation, extra bases must have been added to the primer when constructing the donor plasmid. And this kind of minor sequence modification must be target-specific. If no extra bases were added, the so-called precise KI cannot result in correct eGFP expression. If extra bases were added, the authors must have mentioned it in the text to remind those who are interested in using ORAGNE in the future. 

3. From the previous study of Palmer et al., (Comp Funct Genomics. 2004 Jun; 5(4): 342–353.doi: 10.1002/cfg.405), N-terminal tagging with GFP adversely affects the protein localization in reverse transfection assays, whereas tagging with GFP at the C-terminal is generally better in preserving the localization of the native protein. To further confirmation of the correct expression of eGFP-tagged protein, especially for those with N-tagged (i.e. CADPS, Syt7, CACNA1A, CACNA1E, Grin1, Grin2a and Grin2b in table S1), a specific antibody must be used. The authors should put the evidences for the co-localization of endogenous detected by eGFP and specific antibody, not just mention in the discussion.

Reviewer #2: The authors present a systematic approach of on-locus tagging of neuronal proteins (cultivated hippocampal neurons, slice culture) based on adaptations of CRISPR/Cas9 technology (they name their approach ORANGE). They present successful on locus tagging, and subject the tagged proteins to conventional co-labelling experiments, as well super-resolution and single molecule imaging experiments. 

Let me start with the notion that I am not an expert in the application of CRISPR/Cas9

technology to mammalian neurons. I do feel that their approach is relatively standard but might well be wrong. Concerning the subsequent characterization on chosen examples: this is as far as I can see done largely well, however, certainly pertains only to their chosen examples. Quite honestly, I do not think that stating that endogenously labelled neuronal proteins can per se suffice to retrieve super-resolved or single molecule data (here this least certainly) is particularly surprising. There are quite a few examples of this kind, also from non-mammalian neurobiology. Overall, my enthusiasm for this study is rather moderate as it really seems a rather standard approach. Again, in case my colleagues are of different opinion concerning the novelty and importance of this “resource” this reviewer would not object an overall more positive judgement. 

Specific points: 

- Ensuring XFP-on locus integration certainly is not a per se guarantee for functionality, as the sheer presence of the XFP can certainly be problematic. This from my experience is particularly relevant for membrane proteins such as glutamate receptors. Thus: ultimately to generation of such a resource would need strategies to functionally validate the fusion products. This, at least in my view, must be done by genetic rescue experiments targeting specific phenotypes of the gene under study. This, however, can hardly be part of such a resource approach. 

- Along similar lines: tagging exclusively N- or C-termini will not suffice ultimately to arrive a fully functional versions, particularly for membrane proteins.

- They honestly state: “besides correct integration, we found various insertions and deletions leading to frameshift mutations. We noted that the frequency of indels was highly variable between different knock-ins, which is likely because the accuracy of Cas9 and NHEJ is reported to highly depend on the target sequence (25, 26)”: 

but isnt that potentially dangerous, as e.g. the second allele of a targeted locus could be become unfunctional or worse start to express gain-of-function partial proteins etc? What is the exact frequency of such events compared to “correct” ones? 

- “While this fatefully labels proteins of interest”: Do they really mean that?

- Why is there in Fig. 3E no GFP group?

Reviewer #3: Genome editing through CRISPR/Cas9 has revolutionised biology in recent years. Many laboratories have developed CRISPR/Cas9 based techniques to insert Tag/ORF DNA sequences with great precision at any desired genomic locus in both dividing and postmitotic cells. These technologies are improving rapidly and have proven invaluable in addressing a wide range of questions in research fields such as cancer biology, developmental biology and neuroscience. 

Willems and colleagues offer an original contribution to this expanding repertoire of genome editing tools. They have developed a simple cassette, which they dubbed pORANGE (Open Resource for the Application of Neuronal Genome Editing), that, through a simple two-step cloning procedure can specify any gene/protein for in frame tagging with a GFP. The authors note that the GFP can easily be replaced with any other fluorescent protein or any other tag, greatly expanding the range of experimental approaches one could choose. Further important advantages of the method include: 1) The expression at physiological levels of the tagged proteins as opposed to most traditional overexpression methods. 2) Less likely to interfere with protein localisation and function. 3) Compatible with super resolution of tagged proteins of live as well as fixed tissues.

Additionally, the authors have generated an extensive library of ‘target’ plasmids that allows tagging with GFP of a large number of proteins of interest to researchers interested in synapse biology.

The potential and versatility of the tools are excellently illustrated using the PSD95 and NMDA receptor GFP tagging in primary neurons and hippocampal slice cultures. The authors show that the approach is compatible with different gene delivery systems. Super resolution microscopy, live cell imaging and single molecule localisation microscopy are used to illustrate the proper behaviour of EGF-tagged NMDA receptors. Quantification of fluorescence intensities demonstrate the tagged proteins are expressed at normal physiological level, greatly adding confidence to this approach.

It is envisaged that these tools can easily be adapted for molecules of interest in other aspects of the neurosciences and other fields of cell biological research and as such should appeal to a broad scientific audience.

The authors should indicate how interested colleagues can obtain the plasmids. Will they be deposited with Addgene?

Reviewer #4: In this manuscript, Willems et al. present an interesting strategy and toolkit for genome engineering in order to label endogenous neuronal proteins for optical imaging. The strategy is interesting and offers potential as an alternative to overexpression of fluorescently-tagged proteins. A particularly interesting aspect of this paper is the development of a construct library for a series of target proteins which could represent a useful resource for the community. 

However, the manuscript, as presented, contains a number of pitfalls and limitations that should be addressed, as well as certain demonstrations that fall short of showing the usefulness of their approach. 

1- A major concern with this approach is the presence of indels and frameshift mutations in many cells in which GFP is not expressed. The authors downplay the impact of the partial or complete knockout of the target protein in non fluorescent cells, but this could impact the network signaling, thereby reducing the applicability of this approach. Furthermore, the outcome of the second copy of the gene in GFP positive cells remains unclear. The author suggest that expression levels of PSD95 was “modestly lower” in some GFP positive cells. Could the downregulation be more pronounced for other constructs (no data was shown for this)? If so, then the level of expression of the targeted protein could impact synaptic, cytoskeletal (etc) function. Or could some be mutated ? If so, this will impact function of multimeric proteins. 

2- The applicability of ORANGE would be most relevant in live-cell experiments, without using antibodies or nanobodies. However, the authors show only one live-cell experiment relying on the signal of a transfected protein is a supplementary figure. In fact, while the authors argue in the introduction of fixation and immunolabeling can introduce ultrastructural changes and interfere with protein localization, for the most part, their study uses fixation and/or immunolabeling. 

3- On lines 147-149 the authors state “For none of the knock-in constructs in our library, we observed aberrant or diffuse expression of the integrated tag, indicating that off-target integration, or unintended GFP expression directly from the knock-in plasmid did not occur or is extremely rare”. 

How was aberrant expression assessed for each protein tested (other than PSD95, b3-tubulin and b-actin) ? 

It would be useful to describe any aberrant or diffuse expression for other tested constructs that were not included in this library and if this is more susceptible to happen with particular types of proteins. 

4- Throughout the paper, figures mostly show single cell compartments. It would be useful to show large fields of view to appreciate the level and localization of expression in multiple cells.

5- While the authors tested several proteins that are yet uncharacterized (e.g. c9orf4 and GSG1L), they did not show their cellular distribution using super-resolution. All of the super-resolution experiments were focussed on previously observed proteins, thus reducing the novelty impact of their study.

6- For the super-resolution imaging experiments, the GFP signal was amplified with either nanobodies or antibodies against GFP. This is not always clearly stated in the text and should be standardized throughout the manuscript.

7- In line 339, it is stated that receptor trajectories were “found within the boundaries of the PSD”. This should be rephrased to avoid confusion with PSD95 staining that was used throughout the study, since in this case, as described in the figure legend (Fig7F), the “PSD-mask” was obtained from overexpressed Homer1c-mCherry widefield signal. For a better representation of the synaptic areas, the authors could have used PSD95-FingR co-transfection.

8- Multicolor imaging experiments presented throughout the paper exploit the ORANGE strategy for only one protein, while the other is revealed from either immunostaining or overexpression of fluorescently tagged proteins. Could the ORANGE strategy be applicable to multicolor labeling ? As multicolor imaging is nowadays essential in almost every experimental protocol, they should demonstrate how this can be achieved with the ORANGE strategy.

---

## [Editor Report · Decision Letter 2]

8 Oct 2019

Dear Harold,

Thank you again for your patience whilst we considered your formal appeal. As mentioned previously, we would like to give you the opportunity to revise your manuscript based on the reviewers' comments and your detailed responses to them.

We expect your revision to address the specific points made by each reviewer and be submitted within two months. Keeping this deadline is important for further consideration. 

Please note that we expect you to add additional data, as you have suggested in your detail response to reviewers, including to reviewer 1’s points 1 and 3 and reviewer 4’s points 2, 3, 4, 5, and -importantly- 8, as well as showing efficient labeling of endogenous proteins in vivo in mouse brain, as you indicated in your cover letter.

Please submit a file detailing your responses to the editorial requests and a point-by-point response to all of the reviewers' comments that indicates the changes you have made to the manuscript. In addition to a clean copy of the manuscript, please upload a 'track-changes' version of your manuscript that specifies the edits made. This should be uploaded as a "Related" file type. You should also cite any additional relevant literature that has been published since the original submission and mention any additional citations in your response. 

Before you revise your manuscript, please review the following PLOS policy and formatting requirements checklist PDF: http://journals.plos.org/plosbiology/s/file?id=9411/plos-biology-formatting-checklist.pdf. It is helpful if you format your revision according to our requirements - should your paper subsequently be accepted, this will save time at the acceptance stage.

Please note that as a condition of publication PLOS' data policy (http://journals.plos.org/plosbiology/s/data-availability) requires that you make available all data used to draw the conclusions arrived at in your manuscript. If you have not already done so, you must include any data used in your manuscript either in appropriate repositories, within the body of the manuscript, or as supporting information (N.B. this includes any numerical values that were used to generate graphs, histograms etc.). For an example see here: http://www.plosbiology.org/article/info%3Adoi%2F10.1371%2Fjournal.pbio.1001908#s5.

For manuscripts submitted on or after 1st July 2019, we require the original, uncropped and minimally adjusted images supporting all blot and gel results reported in an article's figures or Supporting Information files. We will require these files before a manuscript can be accepted so please prepare them now, if you have not already uploaded them. Please carefully read our guidelines for how to prepare and upload this data: https://journals.plos.org/plosbiology/s/figures#loc-blot-and-gel-reporting-requirements.

Upon resubmission, the editors will assess your revision and if the editors and Academic Editor feel that the revised manuscript remains appropriate for the journal, we will send the manuscript for re-review. We aim to consult the same Academic Editor and reviewers for revised manuscripts but may consult others if needed.

We expect to receive your revised manuscript within two months. Please email us (plosbiology@plos.org) to discuss this if you have any questions or concerns, or would like to request an extension. At this stage, your manuscript remains formally under active consideration at our journal; please notify us by email if you do not wish to submit a revision and instead wish to pursue publication elsewhere, so that we may end consideration of the manuscript at PLOS Biology.

When you are ready to submit a revised version of your manuscript, please go to https://www.editorialmanager.com/pbiology/ and log in as an Author. Click the link labelled 'Submissions Needing Revision' where you will find your submission record. 

Sincerely,

Gabriel Gasque, Ph.D., 

Senior Editor

PLOS Biology

Reviewer remarks:

Reviewer #1: In general, the authors established a very good tagging toolkit for monitoring the endogenous protein by using HITI-CRISPR techniques for taking the advantages of HITI can work with higher efficiency in post-mitotic cells. With the combination of the high resolution of gSTED microscopy images, ORANGE Toolkit indeed provides a powerful approach to monitor the expression and dynamics in living cells, especially for those proteins without good antibodies.

There are some minor comments regarding this manuscript:

1. For HITI-CRISPR precise KI analysis, it is strongly suggested that only analyze the eGFP-positive cells instead of analyze a pool of cells. A N-tag of eGFP might cause out-of-frame for the coding sequences of target gene, when the other allele is also edited, a KO cell was generated. If the target region is not analyzed in sorted cells (i.e. eGFP positive cells), the aberrant expression pattern cannot be ruled out unless a specific antibody was used parallely to double check the pattern.

2. In Figure S3-D, due to duplication of incomplete target sequences (17 bases, 5’tggcactgggcaggcaggg) in the 5’juntion in GFP-CamKIIa, a new ATG start codon was generated. This new ATG-frame will cause an out-of-frame reading to the subsequent eGFP coding sequence. To overcome this situation, extra bases must have been added to the primer when constructing the donor plasmid. And this kind of minor sequence modification must be target-specific. If no extra bases were added, the so-called precise KI cannot result in correct eGFP expression. If extra bases were added, the authors must have mentioned it in the text to remind those who are interested in using ORAGNE in the future.

3. From the previous study of Palmer et al., (Comp Funct Genomics. 2004 Jun; 5(4): 342–353.doi: 10.1002/cfg.405), N-terminal tagging with GFP adversely affects the protein localization in reverse transfection assays, whereas tagging with GFP at the C-terminal is generally better in preserving the localization of the native protein. To further confirmation of the correct expression of eGFP-tagged protein, especially for those with N-tagged (i.e. CADPS, Syt7, CACNA1A, CACNA1E, Grin1, Grin2a and Grin2b in table S1), a specific antibody must be used. The authors should put the evidences for the co-localization of endogenous detected by eGFP and specific antibody, not just mention in the discussion.

Reviewer #2: The authors present a systematic approach of on-locus tagging of neuronal proteins (cultivated hippocampal neurons, slice culture) based on adaptations of CRISPR/Cas9 technology (they name their approach ORANGE). They present successful on locus tagging, and subject the tagged proteins to conventional co-labelling experiments, as well super-resolution and single molecule imaging experiments.

Let me start with the notion that I am not an expert in the application of CRISPR/Cas9

technology to mammalian neurons. I do feel that their approach is relatively standard but might well be wrong. Concerning the subsequent characterization on chosen examples: this is as far as I can see done largely well, however, certainly pertains only to their chosen examples. Quite honestly, I do not think that stating that endogenously labelled neuronal proteins can per se suffice to retrieve super-resolved or single molecule data (here this least certainly) is particularly surprising. There are quite a few examples of this kind, also from non-mammalian neurobiology. Overall, my enthusiasm for this study is rather moderate as it really seems a rather standard approach. Again, in case my colleagues are of different opinion concerning the novelty and importance of this “resource” this reviewer would not object an overall more positive judgement.

Specific points:

- Ensuring XFP-on locus integration certainly is not a per se guarantee for functionality, as the sheer presence of the XFP can certainly be problematic. This from my experience is particularly relevant for membrane proteins such as glutamate receptors. Thus: ultimately to generation of such a resource would need strategies to functionally validate the fusion products. This, at least in my view, must be done by genetic rescue experiments targeting specific phenotypes of the gene under study. This, however, can hardly be part of such a resource approach.

- Along similar lines: tagging exclusively N- or C-termini will not suffice ultimately to arrive a fully functional versions, particularly for membrane proteins.

- They honestly state: “besides correct integration, we found various insertions and deletions leading to frameshift mutations. We noted that the frequency of indels was highly variable between different knock-ins, which is likely because the accuracy of Cas9 and NHEJ is reported to highly depend on the target sequence (25, 26)”:

but isnt that potentially dangerous, as e.g. the second allele of a targeted locus could be become unfunctional or worse start to express gain-of-function partial proteins etc? What is the exact frequency of such events compared to “correct” ones?

- “While this fatefully labels proteins of interest”: Do they really mean that?

- Why is there in Fig. 3E no GFP group?

Reviewer #3: Genome editing through CRISPR/Cas9 has revolutionised biology in recent years. Many laboratories have developed CRISPR/Cas9 based techniques to insert Tag/ORF DNA sequences with great precision at any desired genomic locus in both dividing and postmitotic cells. These technologies are improving rapidly and have proven invaluable in addressing a wide range of questions in research fields such as cancer biology, developmental biology and neuroscience.

Willems and colleagues offer an original contribution to this expanding repertoire of genome editing tools. They have developed a simple cassette, which they dubbed pORANGE (Open Resource for the Application of Neuronal Genome Editing), that, through a simple two-step cloning procedure can specify any gene/protein for in frame tagging with a GFP. The authors note that the GFP can easily be replaced with any other fluorescent protein or any other tag, greatly expanding the range of experimental approaches one could choose. Further important advantages of the method include: 1) The expression at physiological levels of the tagged proteins as opposed to most traditional overexpression methods. 2) Less likely to interfere with protein localisation and function. 3) Compatible with super resolution of tagged proteins of live as well as fixed tissues.

Additionally, the authors have generated an extensive library of ‘target’ plasmids that allows tagging with GFP of a large number of proteins of interest to researchers interested in synapse biology.

The potential and versatility of the tools are excellently illustrated using the PSD95 and NMDA receptor GFP tagging in primary neurons and hippocampal slice cultures. The authors show that the approach is compatible with different gene delivery systems. Super resolution microscopy, live cell imaging and single molecule localisation microscopy are used to illustrate the proper behaviour of EGF-tagged NMDA receptors. Quantification of fluorescence intensities demonstrate the tagged proteins are expressed at normal physiological level, greatly adding confidence to this approach.

It is envisaged that these tools can easily be adapted for molecules of interest in other aspects of the neurosciences and other fields of cell biological research and as such should appeal to a broad scientific audience.

The authors should indicate how interested colleagues can obtain the plasmids. Will they be deposited with Addgene?

Reviewer #4: In this manuscript, Willems et al. present an interesting strategy and toolkit for genome engineering in order to label endogenous neuronal proteins for optical imaging. The strategy is interesting and offers potential as an alternative to overexpression of fluorescently-tagged proteins. A particularly interesting aspect of this paper is the development of a construct library for a series of target proteins which could represent a useful resource for the community.

However, the manuscript, as presented, contains a number of pitfalls and limitations that should be addressed, as well as certain demonstrations that fall short of showing the usefulness of their approach.

1- A major concern with this approach is the presence of indels and frameshift mutations in many cells in which GFP is not expressed. The authors downplay the impact of the partial or complete knockout of the target protein in non fluorescent cells, but this could impact the network signaling, thereby reducing the applicability of this approach. Furthermore, the outcome of the second copy of the gene in GFP positive cells remains unclear. The author suggest that expression levels of PSD95 was “modestly lower” in some GFP positive cells. Could the downregulation be more pronounced for other constructs (no data was shown for this)? If so, then the level of expression of the targeted protein could impact synaptic, cytoskeletal (etc) function. Or could some be mutated ? If so, this will impact function of multimeric proteins.

2- The applicability of ORANGE would be most relevant in live-cell experiments, without using antibodies or nanobodies. However, the authors show only one live-cell experiment relying on the signal of a transfected protein is a supplementary figure. In fact, while the authors argue in the introduction of fixation and immunolabeling can introduce ultrastructural changes and interfere with protein localization, for the most part, their study uses fixation and/or immunolabeling.

3- On lines 147-149 the authors state “For none of the knock-in constructs in our library, we observed aberrant or diffuse expression of the integrated tag, indicating that off-target integration, or unintended GFP expression directly from the knock-in plasmid did not occur or is extremely rare”.

How was aberrant expression assessed for each protein tested (other than PSD95, b3-tubulin and b-actin) ?

It would be useful to describe any aberrant or diffuse expression for other tested constructs that were not included in this library and if this is more susceptible to happen with particular types of proteins.

4- Throughout the paper, figures mostly show single cell compartments. It would be useful to show large fields of view to appreciate the level and localization of expression in multiple cells.

5- While the authors tested several proteins that are yet uncharacterized (e.g. c9orf4 and GSG1L), they did not show their cellular distribution using super-resolution. All of the super-resolution experiments were focussed on previously observed proteins, thus reducing the novelty impact of their study.

6- For the super-resolution imaging experiments, the GFP signal was amplified with either nanobodies or antibodies against GFP. This is not always clearly stated in the text and should be standardized throughout the manuscript.

7- In line 339, it is stated that receptor trajectories were “found within the boundaries of the PSD”. This should be rephrased to avoid confusion with PSD95 staining that was used throughout the study, since in this case, as described in the figure legend (Fig7F), the “PSD-mask” was obtained from overexpressed Homer1c-mCherry widefield signal. For a better representation of the synaptic areas, the authors could have used PSD95-FingR co-transfection.

8- Multicolor imaging experiments presented throughout the paper exploit the ORANGE strategy for only one protein, while the other is revealed from either immunostaining or overexpression of fluorescently tagged proteins. Could the ORANGE strategy be applicable to multicolor labeling ? As multicolor imaging is nowadays essential in almost every experimental protocol, they should demonstrate how this can be achieved with the ORANGE strategy.

---

## [Decision Letter · Decision Letter 3]

28 Jan 2020

Dear Harold,

Thank you for submitting your revised Methods and Resources entitled "ORANGE: A CRISPR/Cas9-based genome editing toolbox for epitope tagging of endogenous proteins in neurons" for publication in PLOS Biology. I have now obtained advice from the original reviewers 1, 2, and 4, and have discussed their comments with the Academic Editor. 

We're delighted to let you know that we're now editorially satisfied with your manuscript. However before we can formally accept your paper and consider it "in press", we also need to ensure that your article conforms to our guidelines. A member of our team will be in touch shortly with a set of requests. As we can't proceed until these requirements are met, your swift response will help prevent delays to publication. Please also make sure to address the data and other policy-related requests noted at the end of this email.

*Copyediting*

*Published Peer Review History*

*Early Version*

*Submitting Your Revision*

Sincerely,

Gabriel Gasque, Ph.D., 

Senior Editor

PLOS Biology

ETHICS STATEMENT:

The Ethics Statements in the submission form and Methods section of your manuscript should match verbatim. Please ensure that any changes are made to both versions.

-- Please include the ID number(s) of the protocol(s) approved by the Dutch Animal Experiments Committee.

DATA POLICY:

Note that we do not require all raw data. Rather, we ask for all individual quantitative observations that underlie the data summarized in the figures and results of your paper. For an example see here: http://www.plosbiology.org/article/info%3Adoi%2F10.1371%2Fjournal.pbio.1001908#s5.

These data can be made available in one of the following forms:

Regardless of the method selected, please ensure that you provide the individual numerical values that underlie the summary data displayed in the following figure panels: Figure 4B-E, 4G, 4I, 5BC, 5EF, 7B, 7E-H, 7L, S5D, S6C-F, S7D, S7H, S8B, S8D, S8F, S9F, and S10D-F.

Please also ensure that the figure legends in your manuscript include information on where the underlying data can be found, and ensure your supplemental data file/s has a legend.

Please cite Table S5 within your manuscript.

Reviewer remarks:

Reviewer #1: For the major concern regarding the tag-KI to one allele of the target locus and KO of the other allele, the authors have shown the identical cellular localization of the proteins with double confirming ways by either observing GFP-tagged protein and antibody-stained protein or double confirm the localization of N-, and C-tagged protein. Although just for the proteins studied in this paper, the results are pretty convincing to me. 

The authors demonstrated a great comparison of the KI efficiencies from different gRNAs identified from different platforms, such as MIT, Doench, Bae scores (FigS7). I'd like to pinpoint this for researchers using CRISPR/Cas9 in their study, the results might suggest never try the target a locus with a single gRNA chosen from a single platform. Additionally, the NGS analysis for indel of the target regions in various genes provides important info for gRNA editing profiling, it can be seen from this that choosing different gRNAs target same locus is an important issue in CRISPR/Cas9-mediated KI.

In summary, the authors have proven the ORANGE system can be used as a good tool to tag endogenous protein in post-mitotic cell, neuron in this case. I think it's suitable for PLOS Biology for publication.

Reviewer #2: I did read their response and revised manuscript with interest. I am in oarts not really convinced with their responses, and still with my opinion that the novelty of the approach is not quite adequate 

with the standard of Plos Biology despite the enormous effort associated with their work. However, as said before, this reviewer would not object an overall more

positive judgement of the colleagues. I therefore do not formulate any suggestion concerning acceptance. 

Reviewer #4: The revised manuscript provides a lot of new data that makes this study much more convincing. Overall the study provides very nice tools that will be very useful to the community. The authors have done a great job including the clear acknowledgement of the limitation of the approach.

---

## [Editor Report · Decision Letter 4]

27 Mar 2020

Dear Dr MacGillavry,

On behalf of my colleagues and the Academic Editor, Yi-Ping Hsueh, I am pleased to inform you that we will be delighted to publish your Methods and Resources in PLOS Biology. 

Early Version

PRESS 

Kind regards,

Alice Musson

Publication Assistant, 

PLOS Biology

on behalf of

Gabriel Gasque,

Senior Editor

PLOS Biology